# Understanding the charge transfer effects of single atoms for boosting the performance of Na-S batteries

Yao-Jie Lei[1,2,8], Xinxin Lu [1,8], Hirofumi Yoshikawa [3], Daiju Matsumura[3], Yameng Fan[1], Lingfei Zhao[1], Jiayang Li[1], Shijian Wang [2], Qinfen Gu[4], Hua-Kun Liu [5], Shi-Xue Dou [5], Shanmukaraj Devaraj [6], Teofilo Rojo[7], Wei-Hong Lai [1] ✉, Michel Armand [6] ✉, Yun-Xiao Wang[1,5] ✉ & Guoxiu Wang [2] ✉

The effective flow of electrons through bulk electrodes is crucial for achieving high-performance batteries, although the poor conductivity of homocyclic sulfur molecules results in high barriers against the passage of electrons through electrode structures. This phenomenon causes incomplete reactions and the formation of metastable products. To enhance the performance of the electrode, it is important to place substitutable electrification units to accelerate the cleavage of sulfur molecules and increase the selectivity of stable products during charging and discharging. Herein, we develop a single-atom-charging strategy to address the electron transport issues in bulk sulfur electrodes. The establishment of the synergistic interaction between the adsorption model and electronic transfer helps us achieve a high level of selectivity towards the desirable short-chain sodium polysulfides during the practical battery test. These finding indicates that the atomic manganese sites have an enhanced ability to capture and donate electrons. Additionally, the charge transfer process facilitates the rearrangement of sodium ions, thereby accelerating the kinetics of the sodium ions through the electrostatic force. These combined effects improve pathway selectivity and conversion to stable products during the redox process, leading to superior electrochemical performance for room temperature sodium-sulfur batteries.

Lithium-ion batteries have established themselves as the primary option for powering portable electronic devices and electric vehicles[1–3]. The limited availability and high price of Li, however, have led to the increased popularity of sodium-based batteries, owing to the low cost and natural abundance of sodium resources. Among these sodium-based storage technologies, room temperature sodium-sulfur (RT Na-S) batteries are particularly promising due to their high energy density, up to 1274 Wh·kg$^{-1}$[4–8]. Although progress has been made in developing the Na-S battery, several obstacles remain[9–12]. One

significant challenge is its complex and uncontrollable cathodic products, such as long-chain and short-chain sodium polysulfides (NaPSs) in the RT Na-S system[13–15]. The long-chain (LC) NaPSs (Na$_2$S$_x$, $4 \le x \le 8$) are metastable intermediates. These are highly soluble in ether-based electrolytes or can undergo nucleophilic reactions with carbonate-based electrolytes[11,16–19]. This solubility causes active material loss and interfacial deterioration, which subsequently leads to rapid capacity decay[8,20–24]. In order to overcome these obstacles, advanced catalysts must be developed to effectively regulate the reaction pathways and

A full list of affiliations appears at the end of the paper. ✉e-mail: weihongl@uow.edu.au; marmand@cicenergigune.com; yunxiaowang@usst.edu.cn; Guoxiu.Wang@uts.edu.au

achieve optimal product selectivity during charging and discharging. Product-selectivity, a concept often employed in connection with organic catalysis such as in $CO_2$ reduction[25–27], hydrogenation reactions[28–30], and toluene oxidation[31–33], has rarely been explored in the context of Na-S battery technology. By selectively reducing the production of unstable LC NaPS products, the yield of stable short-chain products can be enhanced, although the reaction kinetics of short-chain products, including $Na_2S_2$ and $Na_2S$ intermediates, is sluggish. The continuous build-up of solid NaPSs on the electrode reduces its charge transfer capability and obstructs ion accessibility, potentially causing severe polarization, irreversible capacity, and poor Coulombic efficiency[7,34–37]. Despite numerous studies focusing on improving the reactivity and stability of RT Na-S batteries by incorporating catalysts into cathodes, the diverse catalytic activities exhibited by various catalysts towards sulfur and sodium-sulfide species have not been thoroughly investigated[38–40]. Altering the catalytic active sites can have a significant impact on the composition of reaction products during discharge and charge processes, since catalysts with different electronic structures exhibit varying electron transfer (ET) capabilities, which, in turn, influence reaction pathways and product-selectivity. Thus, these challenges necessitate building and scaling catalytic relationships between catalysts and products to achieve optimal interfacial properties and enhance their reversibility under reactions within the batteries.

Understanding and quantifying product-selectivity is crucial, as it can inspire new approaches and directions for the use of catalysts in battery technology. In this study, we first construct the collaborative relationship between the absorption model and ET. This method can help us rapidly screen out promising single atom catalysts that have a high level of selectivity towards the short-chain sodium polysulfides for sodium sulfur batteries. This allows for understanding the relationships between product selectivity and the electrification capability

of single atom catalysts (SACs) for Na-S batteries. Based on previous catalytic work theory, we believe that establishing a weak intermolecular bonding force with fast ET capability is necessary to achieve a low energy barrier for sulfur dissociation. The rapid cleavage of sulfur molecules can avoid the formation of unstable sodium polysulfides, resulting in enhanced selectivity towards stable short chain polysulfides. To confirm these predictions, we experimentally investigated the effects of different single atoms on the reaction pathways and product selectivity of RT Na-S batteries by fabricating a series of single-atom metal catalysts supported on porous nitrogen-doped carbon nanospheres ($M_1$-PNC). The catalysts included manganese ($Mn_1$), iron ($Fe_1$), cobalt ($Co_1$), tin ($Sn_1$), nickel ($Ni_1$), and copper ($Cu_1$), with each showing distinct ET capabilities. With its stronger ability to donate electrons, $Mn_1$ promotes the decomposition of polysulfide chains, leading to the rapid formation of short-chain NaPSs and $Na_2S$. Additionally, $Mn_1$ with its strong ability to capture electrons reduces the irreversibility of solid-state short-chain NaPSs during the charging process, facilitating the conversion of $Na_2S$ to NaPS. Consequently, the reversible capacity and stability of RT Na-S batteries are enhanced. $Mn_1$ active sites exhibit high product-selectivity towards short-chain NaPSs among these six metals ($Mn_1$, $Fe_1$, $Co_1$, $Sn_1$, $Ni_1$, $Cu_1$), which suppresses the "shuttle effect" and substitution reactions with electrolytes, promoting cycling stability in RT Na-S batteries.

## Results

### Screening single atom catalysts

Generally, reactions on heterogeneous interfaces involve adsorption and desorption processes. According to the Sabatier principle, if the adsorption of reactants is too strong, the reaction rate depends mainly on the desorption energy, while if it is too weak, the reaction rate depends mainly on the adsorption energy (Fig. 1a)[41]. Both scenarios can lead to poor reaction rates. In the case of sodium-sulfur batteries,

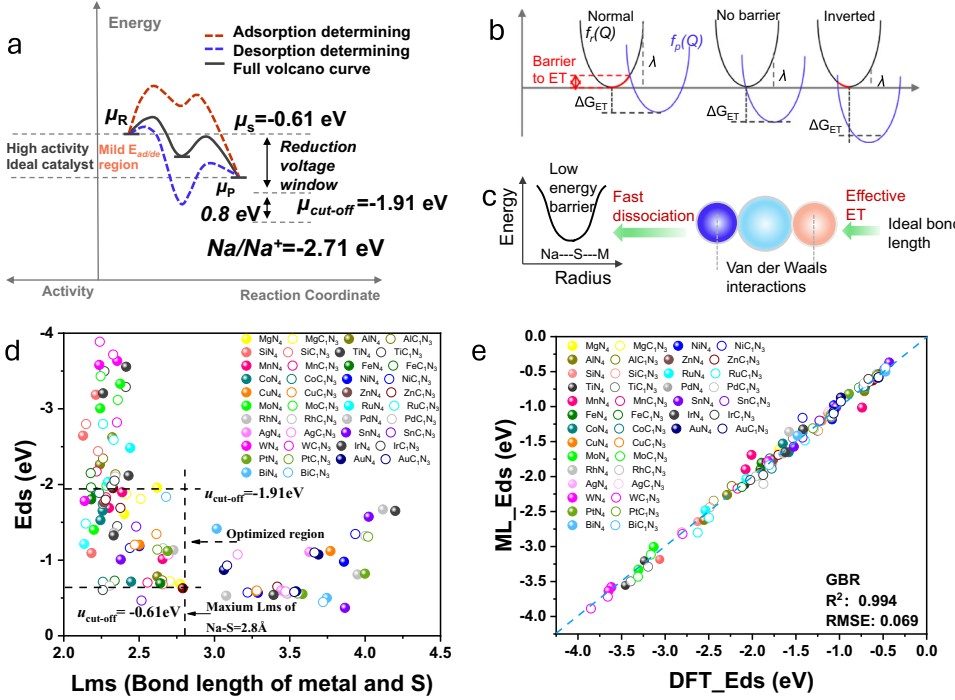

**Fig. 1 | Screen single atom catalysts. a** Schematic diagram of the typical curves associated with reactions in which the adsorption (red) and desorption (blue) are rate determining step s, together with the key potentials in Na-S batteries (right). **b** Reactant ($f_r(Q)$) and product ($f_p(Q)$) potential energy wells for electron transfer in three different regimes: weak/ideal/strong electron coupling (from left to right). **c** Diagram of the creation of sulfur-bonded heteroatoms as a function of the

distance between the sulfur and the metal atoms, together with a schematic diagram of ideal van der Waals interactions to achieve low-energy-barrier bonds. **d** Correlations between the adsorption energy ($E_{ds}$) and the bond length between the metal and S ($L_{ms}$). **e** Comparison of $E_{ds}$ values from the DFT calculations and the full-fit results using the ML algorithm for gradient boosted regression (GBR).

the theoretical reduction potential of the reactant sulfur is −0.61 eV (versus reversible hydrogen electrode (RHE))[42,43]. The onset redox potentials of $Na_2S_6$ is around 2.1 V (versus $Na/Na^+$), which corresponds to −0.61 V (versus RHE). Also, the cutoff discharge voltage is 0.8 V (versus $Na/Na^+$), which corresponds to −1.91 V (versus RHE). Based on these conditions for the battery test, it can be concluded that the cut-off chemical potential for all products obtained at the cathode is −0.61 eV to −1.91 eV (versus RHE)[37,44-47]. If sulfur needs to be rapidly converted to short-chain (SC) sodium polysulfides (NaPSs), the inter-actions between the catalysts and the cathodic products should not be too strong or too weak within this potential window (−1.91 eV <μ< −0.61 eV, where μ is the chemical potential). To achieve high product selectivity towards SC NaPSs, an ideal solution for the cata-lysts is to transfer electrons from the catalysts to sulfur/sulfide species within the voltage range whilst implementing mild adsorption. This could lead to the fast breakdown of the sulfur into SC NaPSs.

Nevertheless, rapid ET is challenging to achieve, given that the rate of this process depends on the reaction free energy, the re-organization energy linked to the ET, and the electronic coupling between the donor and acceptor, according to Marcus's theory (Eq. (1))[48-50]:

$$k_{ET} = \sqrt{\frac{\pi}{h^2 \cdot \lambda \cdot k_B \cdot T}} \cdot H_{(DA)}^2 \cdot \exp\left(-\frac{\left(\lambda + \triangle G_{ET}^0\right)^2}{4 \cdot \lambda \cdot k_B \cdot T}\right). \quad (1)$$

where $k_{ET}$ is the ET rate, $h$ is Planck's constant, $\lambda$ is the total reorgani-zation energy, $k_B$ is Boltzmann's constant. $T$ is the temperature, $H_{(DA)}$ is the electronic coupling matrix element between donor and acceptor, and $\triangle G_{ET}^0$ is the standard Gibbs' energy change accompanying the ET reaction. Excessively strong or weak re-organization energy often results in high energy barriers for the ET process (Fig. 1b). The re-organization energy depends on the distance between the donor and acceptor ($r_{DA}$) and the solvent polarity (Eq. (2))[51-53]:

$$H_{DA}(r_{DA}) = H_{DA}^{(0)} \cdot \exp\left(-\beta_{el} \cdot \left(r_{DA} - r_{DA}^{(0)}\right)\right). \quad (2)$$

Previous research also suggests that the bond barrier usually has the lowest energy when the bond length typically reaches the van der Waals radius of the atoms involved (Fig. 1c)[54]. Next, machine learning (ML) is performed as a sufficient way to identify the best scientific descriptors, which can assess how various physical and chemical properties of metal atoms affect adsorption or reaction energy. As shown in Fig. S1, it is obviously that the bond length of metal and sulfur can influence the adsorption and reaction energy. According to the prediction in Fig. 1e, a linear relationship between the adsorption energy $E_{ads}$ and diverse SACs is obtained, which demonstrates that the prediction results of ML are like those of DFT calculations. Then, by establishing a linear relationship between the lengths of adsorption and metal-sulfur bonds (Fig. 1d), combined with the theoretical knowledge discussion, we can rapidly screen potential SACs for Na-S batteries that should exist in a specific region (as shown as Fig. 1d, optimized region), in which the SACs, $Mn-N_4$, $Fe-N_4$, $Rh-N_4$, $Mg-N_4$, $Co-N_4$ and $Mg-C_1N_3$ feature mild adsorption (more detailed ML proce-dures are in the Supplementary information, Figs. S2, S3, S4 and Tables S2, S3, S4, S5). With the investigation of prediction, we selected six representative SACs, $Mn_1$, $Fe_1$, $Co_1$, $Sn_1$, $Cu_1$, and $Ni_1$, which were used to conduct further experimental validation in Na-S batteries.

## Synthesis and characterizations of single atom catalysts
These SACs, $Mn_1$, $Fe_1$, $Co_1$, $Sn_1$, $Cu_1$, and $Ni_1$, were synthesized through in-situ deposition on porous nitrogen-doped carbon nanospheres ($M_1$-PNC) via facile polymerization and then carbonization. Afterward, the S was infused into the nanospheres (S@$M_1$-PNC), which then served as cathode materials for Na-S batteries. The scanning electron

microscopy (SEM) and scanning transmission electron microscopy (STEM) images confirm that the S@$Mn_1$-PNC consists of nanospheres with an average diameter of about 500 nm (Supplementary informa-tion, Fig. S5). High-angle annular dark-field (HAADF) STEM and the corresponding intensity profiles further visualize the Mn atoms, which are isolated single atoms dispersed on the nanospheres (Fig. 2a, b). The S@$Mn_1$-PNC was also placed in a field ion microscope (FIM) to be characterized by atom probe tomography (APT) (Fig. 2c). The results demonstrate that C, N, S, and Mn are well-dispersed throughout the nanospheres (Fig. 2d, Supplementary Information, Fig. S6), which is consistent with the energy-dispersive X-ray (EDX) mapping analysis (Supplementary Information, Fig. S7). For comparison, a sample with S loading on $Ni_1$-PNC (S@$Ni_1$-PNC) was analyzed by SEM, STEM, and EDX mapping analysis (Supplementary Information, Figs. S8, S9), confirm-ing that C, N, S, Ni are well-dispersed on the nanosphere with a dia-meter of ~500 nm. Furthermore, the presence of overlapped APT atomic mapping suggests that the atomic distribution of Mn remains unchanged even after loading the S (Fig. 2e).

To further confirm the state of Mn on the nanospheres, X-ray absorption fine structure (XAFS) measurements were employed. The Mn K-edge X-ray absorption near-edge structure (XANES) spectra of S@$Mn_1$-PNC were collected to reveal the oxidation state of Mn. As shown in Fig. S10a (Supplementary Information), the near-edge absorption energy of S@$Mn_1$-PNC almost overlaps with MnO, indi-cating that the Mn atoms have a valence of +2. The Fourier-transform (FT) of the $k^3$-weighted extended X-ray absorption fine structure (EXAFS) spectrum of S@$Mn_1$-PNC shows only one main peak at 1.65 Å (Supplementary Information, Fig. S10b), which can be indexed to the Mn-N coordination shell. The fitting curves and the table of parameters (Supplementary Information, Fig. S10c and Table S6) demonstrate that the $Mn_1$ are coordinated by four nitrogen atoms (and a model was constructed to display this proposed structure). To further verify other single atom metals, we systematically characterized $Fe_1$, $Co_1$, $Sn_1$, $Cu_1$, and $Ni_1$ deposited on carbon nanospheres by HAADF-STEM and XAFS (Fig. 2f and Supplementary Information, Figs. S11, S12, S13, Tables S7, S8).

## Pathway selectivity and electrochemical performance
The S contents in different matrices for S@$M_1$-PNC were measured by thermogravimetric analysis (TGA) (Supplementary Information, Fig. S14). The electrochemical testing was carried out within the vol-tage range of 0.8–2.8 V. The initial discharge and charge (Fig. 3a, Fig. 3b, and Supplementary information Fig. S15) curves for standard cells with various SACs in their cathodes display distinct capacities. Generally, the quasi-solid conversion reaction of $S_8$ cleavages can be divided into two regions in an ester-based electrolyte, including high-plateau capacity and low-plateau capacity. The capacities of these two plateaus are mainly contributed by long-chain (LC) NaPS ($S_8 \rightarrow Na_2S_x \rightarrow Na_2S_4$, 2.8–1.25 V) and short-chain (SC) NaPS ($Na_2S_4 \rightarrow Na_2S_2$, $Na_2S$, 1.25–0.8 V) conversions, respectively. When $Mn_1$ acts as a catalyst for Na-S batteries, the discharge capacity primarily originates from low-plateau capacity, which indicates the capacity are mainly contributed by SC NaPS conversion, accounting for 88.6% of the total capacity (Fig. 3c). In contrast, the high-plateau capacity region contributes 18.3% to the overall capacity. Such result indicates that the $Mn_1$ exhibits high pathway selectivity towards SC NaPS formation. Among the other five SACs, $Fe_1$ possesses similar pathway selectivity to $Mn_1$, in which the capacity contribution of the low-plateau region accounts for 81.7% of the total discharge capability. In the cases of the $Co_1$ and $Sn_1$, the SACs delivered 67% and 64% of low-plateau capacity, respectively. For $Ni_1$ and $Cu_1$, it is evident that the high-plateau capacity ($S \rightarrow Na_2S_6$) region is dominant, suggesting that LC NaPS are main capacity contributor. Since the sluggish kinetics of SC conversion (low-plateau capacity) is the rate-determining step of the S redox reaction, it can be expected that $Mn_1$ would show the best performance, followed

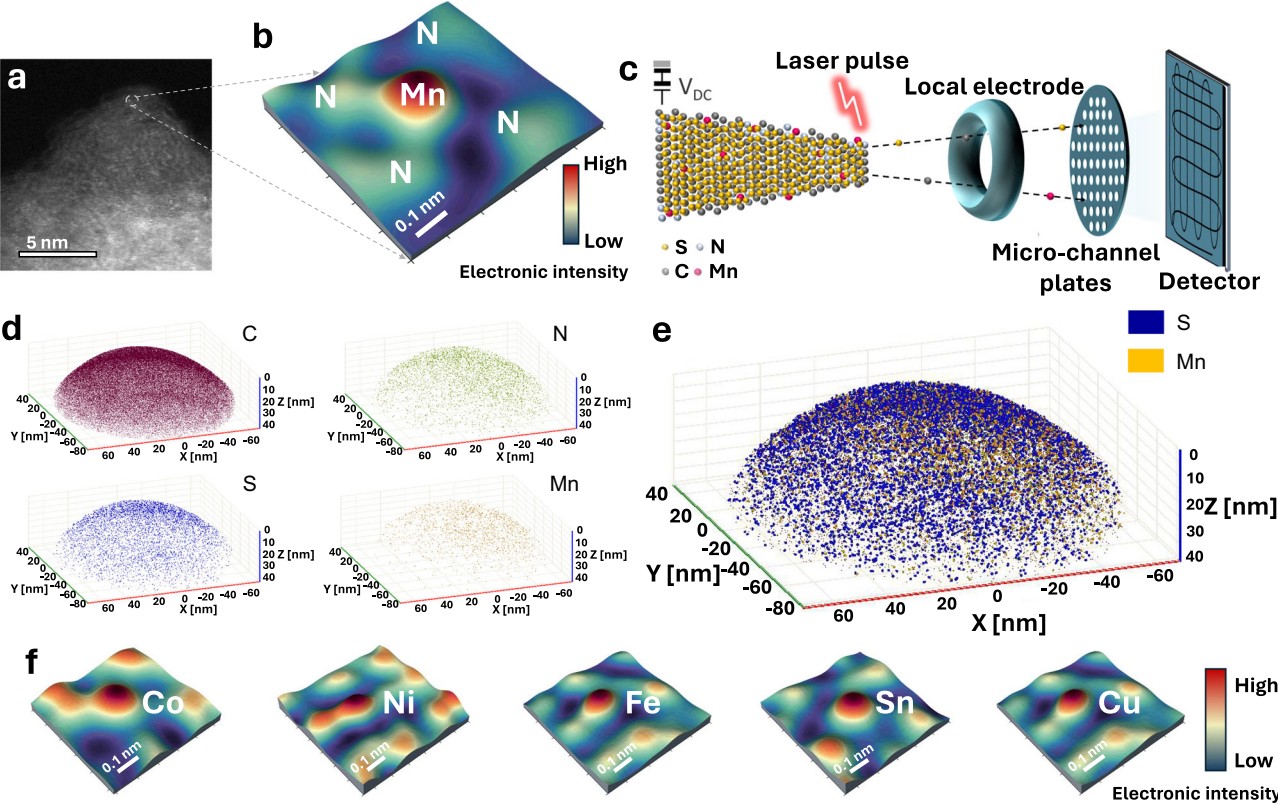

**Fig. 2 | Characterization of single atom catalysts. a** High-resolution HAADF image of $Mn_1@PNC$, together with the three-dimensional (3D) profile of the local structure of the $Mn_1$ atoms (**b**). **c** Schematic illustration of the working mechanism of atomic probe tomography. **d** 3D atomic distribution of the elements: C, N, S, and Mn, and (**e**) their corresponding overlapped 3D atomic distribution map. (**f**) 3D profiles of the local structures of the $Co_1$, $Ni_1$, $Fe_1$, $Sn_1$, and $Cu_1$ atoms.

by $Fe_1$, $Co_1$, $Sn_1$, and finally $Cu_1$ and $Ni_1$. After 120 cycles at a current density of $0.2\,A\,g^{-1}$, the $S@Mn_1$-PNC cathode retains the highest capacity of $784.6\,mAh\,g^{-1}$ amongst all the $S@Mn_1$-PNC samples (Fig. 3d), which also represents the highest capacity retention of 84%.

For comparison, the capacity retention of the $Fe_1$, $Co_1$, $Sn_1$, $Ni_1$ and $Cu_1$ was 77.6%, 65.3%, 62.8%, 55%, and 53%, respectively (Supplementary Information, Fig. S16). Interestingly, the cycling stability of the S@SACs-PNC cathodes was also determined by their pathway selectivity (Supplementary Information, Fig. S17, and Table S9). In addition, $S@Mn_1$-PNC outperformed other S cathodes in terms of rate performance, delivering reversible capacities of 989, 901, 814, 776, 678, and 576 $mAh\,g^{-1}$ at current densities of 0.1, 0.2, 0.5, 1, 2, and $5\,A\,g^{-1}$, respectively (Fig. 3e). When the current density reversed back to $0.1\,A\,g^{-1}$, the $S@Mn_1$-PNC cathode recovered to $890\,mAh\,g^{-1}$, surpassing the other SACs (Supplementary Information, Fig. S18). Compared with the results reported previously, the $S@Mn_1$-PNC cathode exhibited exceptional high-rate capability (Supplementary Information, Fig. S19). Owing to the high-pathway selectivity towards SC NaPSs, the reaction kinetics of $S@Mn_1$-PNC cathode was significantly boosted, which enables excellent cycling stability and superior rate performance. Furthermore, it even displayed excellent long cycling performance, achieving a high reversible capacity of $344.1\,mAh\,g^{-1}$ at $2\,A\,g^{-1}$ after 3000 cycles (Fig. 3f). In additions, Fig. S20 (Supplementary Information) showcases a comparative analysis of long-cycling performance between this work and the previous reports. Due to its fast reaction kinetics, the $S@Mn_1$-PNC remarkably demonstrates outstanding cycling performance for room temperature Na-S batteries.

Overall, single atom catalysts exhibit distinct electrocatalytic activities towards the high-plateau and low-plateau conversion regions, showing different pathway selectivity. Meanwhile, the $S@Mn_1$-PNC and $S@Fe_1$-PNC cathodes demonstrate high reaction kinetics and excellent battery performance via effectively catalyzing the rate-determining step. In contrast, the S cathodes with SACs that tend towards LC selectivity cannot realize good electrochemical performance, because the S redox reactions produce metastable S intermediates (LC NaPSs). Meanwhile, these SACs cause incomplete and highly irreversible final products ($Na_2S$), thus causing rapid active S loss and battery failure.

## Pathway selectivity and cycling stability mechanism

To experimentally explore the product selectivity, which is the pathway selectivity, for $S@Mn_1$-PNC cathode in room temperature (RT) Na-S batteries, the states of sulfur (S) were examined via in-situ synchrotron powder X-ray diffraction (XRD) patterns ($\lambda = 0.6884\,Å$ for $S@Mn_1$-PNC) during the initial discharge and charge processes (Fig. 4a). Under Mn single atom catalysis, S with a typical diffraction peak at 24.4° is directly reduced to form $Na_2S_4$ (23.7°), which is then further reduced to $Na_2S_2$ (27.39°) and $Na_2S$ (38.96°) during the discharge process. Notably, no LC NaPSs were detected when $S@Mn_1$-PNC was used as the cathode. In comparison, when using $Ni_1$, S (Supplementary information, Fig. S21, $\lambda = 0.7748\,Å$) is initially reduced to soluble LC $Na_2S_x$, then converted to $Na_2S_4$ and eventually to $Na_2S_2$ and $Na_2S$ during discharging. Interestingly, the peak of $Na_2S$ disappears gradually during the charging process when using $S@Mn_1$-PNC as the cathode. In contrast, the irreversible $Na_2S$ products remain for the $S@Ni_1$-PNC cathode. These results clearly demonstrate that Mn single atoms exhibit high product-selectivity toward SC NaPSs in the discharge process and excellent ability to promote the reversibility of solid-state SC NaPSs in the charge process. As shown in Fig. 4b, ex-situ X-ray absorption spectroscopy (XAS) measurements for $S@Mn_1$-PNC demonstrate the appearance and disappearance of S K-edge XANES absorption edges in the range of 2465–2480 eV during the initial cycle, further verifying the

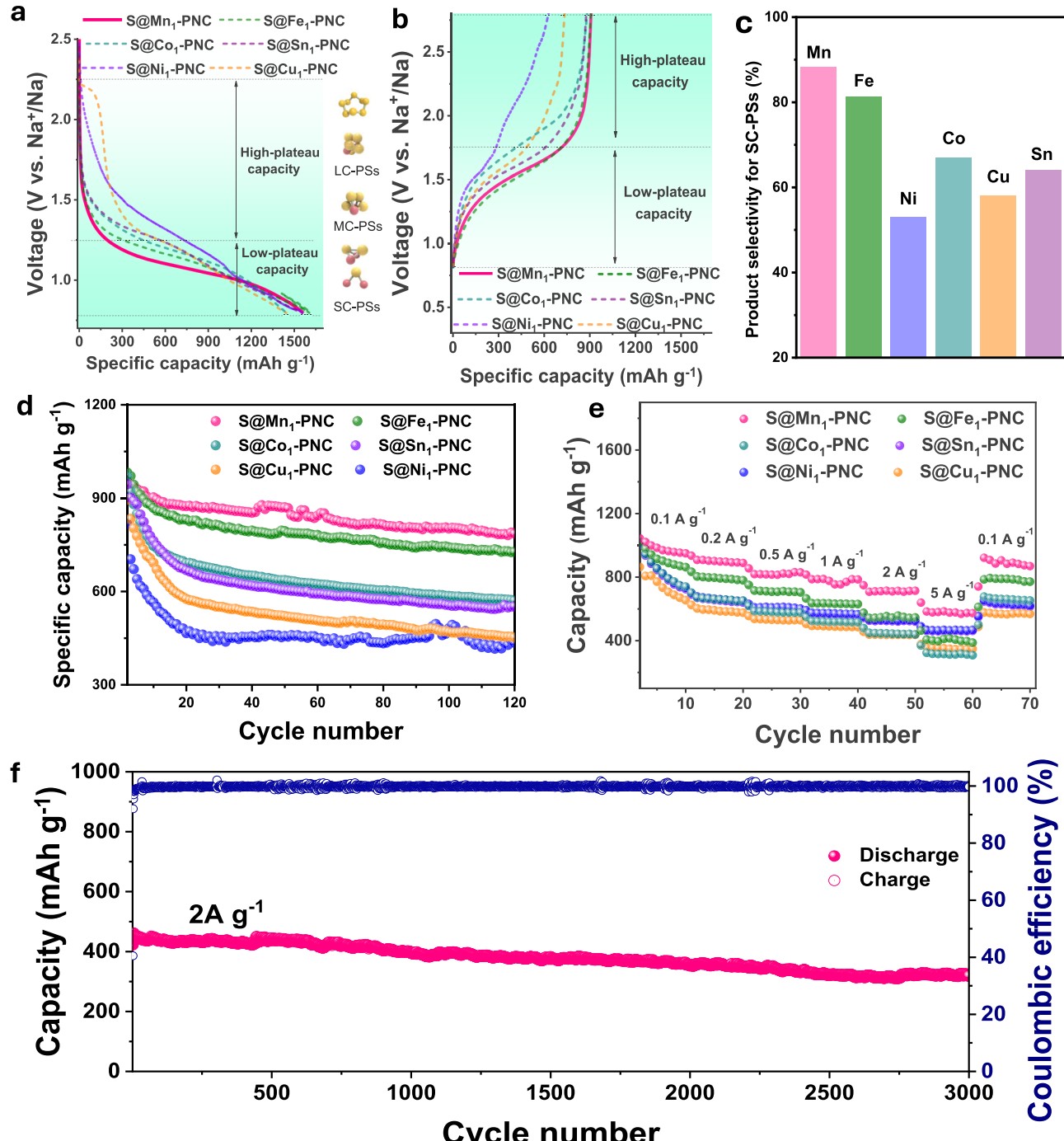

**Fig. 3 | Product selectivity and electrochemical performance for S@Mn₁-PNC, S@Fe₁-PNC, S@Co₁-PNC, S@Sn₁-PNC, S@Ni₁-PNC, and S@Cu₁-PNC. a** Initial discharge and (**b**) charge profiles. **c** The percentages of short-chain polysulfides in the initial discharges. **d** Cycling performances at current density of 0.2 mA g⁻¹. **e** Rate performances. **f** Long-term cycling performance of S@Mn₁-PNC at the current density of 2 A g⁻¹.

conversion of S states during the discharge/charge process and revealing the product-selectivity of Mn₁ towards S intermediates. In comparison, the ex-situ S K-edge XANES spectra of S@Ni₁-PNC (Supplementary Information, Fig. S22) show the conversion process from LC soluble NaPSs to SC insoluble NaPSs during the discharging process. Combining the in-situ XRD with the ex-situ XAS results, Mn single atoms can efficiently reduce polysulfide S₈ to SC sulfide compounds (i.e., Na₂S and Na₂S₂) via Na₂S₄ as dominant intermediate. These results are consistent with the cyclic voltammetry (CV) curves (Supplementary Information, Fig. S23). Time-of-flight secondary-ion mass

spectrometry (TOF-SIMS) was conducted to visually reveal the abundance of both Na and S in the S@Mn₁-PNC electrode after 50 cycles at full discharge to 0.8 V. The 3D reconstructions of the TOF-SIMS depth profiles (Fig. 4c, d) show that both Na and S are uniformly distributed, suggesting the excellent cycling stability of the S@Mn₁-PNC cathode. Normalized depth profiles of secondary ion fragments of bulk S@Mn₁-PNC electrode indicate uniform distributions of both Na and S (Fig. 4e). Furthermore, the top view images of the S@Mn₁-PNC cathode exactly display the evolution of the distributions of both Na and S in different frames, including whole frames, 0–20 frames, 20–40 frames,

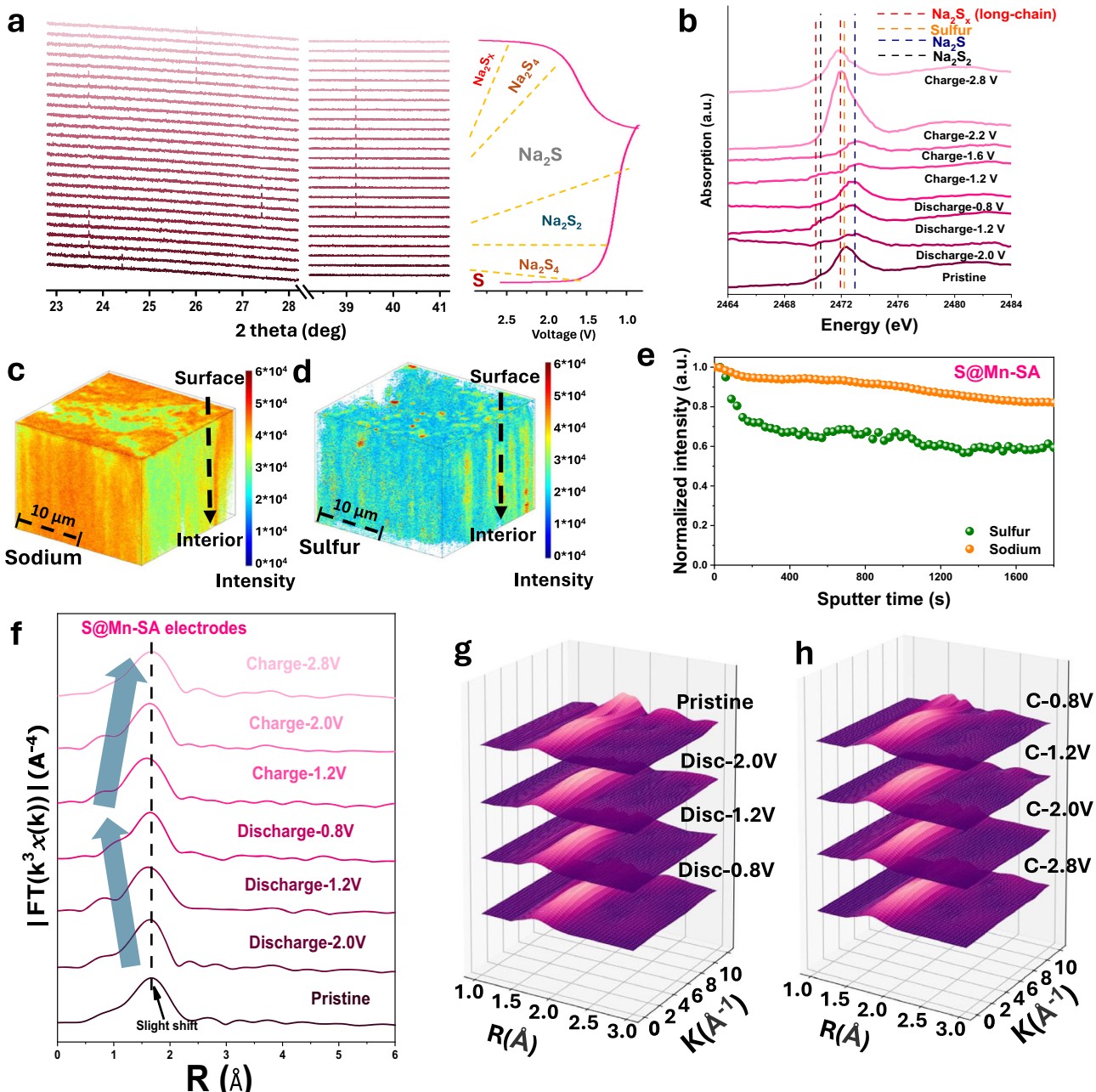

**Fig. 4 | Pathway selectivity and cycling stability mechanism. a** In-situ synchrotron-based XRD patterns of S@Mn$_1$-PNC. **b** Ex-situ X-ray absorption spectra of S for S@Mn$_1$-PNC during the initial cycle. **c** and (**d**) 3D reconstructed images of TOF-SIMS depth profiles of Na and S after 10 cycles. Scale bar 10 μm. **e** Normalized depth profiles of secondary ion fragments obtained from the S@Mn$_1$-PNC electrode after 50 cycles. **f** $k^3$-weighted FT-EXAFS curves of S@Mn$_1$-PNC in R-space during the initial discharge and charge processes. WT plots of the Mn $k$-edge of S@Mn$_1$-PNC cathode during (**g**) discharge and (**h**) charge processes.

and 40−60 frames, respectively, which are consistent with the 3D distributions (Supplementary Information, Fig. S24).

As illustrated by the Fourier-transformed (FT) $k^3$-weighted extended X-ray absorption fine structure (EXAFS) spectrum (Fig. 4f), there is only one primary peak at different voltages, at around 1.65 Å, suggesting that Mn$_1$ single atoms always coordinate with N atoms during the cycling. Interestingly, the peak at 1.65 Å incurs a slight shift to the left during the discharging process, which suggests a decrease in the coordination value of Mn. The result can be attributed to the catalytic effect of SA-Mn$_1$ to S species, which is also consistent with ML prediction. This indicates that $E_{ads}$ will decrease with bond length of Mn-N decreases (Fig. 1d), which facilitates the desorption of S species.

Subsequently, this primary peak recovered its original place during the charging process. Additionally, the continuous wavelet transforms (WTs) of S@Mn$_1$-PNC at various discharge and charge states (Fig. 4g, h) display consistent intensity maxima with no energy changes and values approximately equal to 4.0 Å$^{-1}$, which are all very close to that in the reference Mn-PNC (~4.0 Å$^{-1}$)[55]. This further demonstrates that S@Mn$_1$-PNC has a stable structure, which is advantageous for maintaining cycling stability.

## Charge transfer accelerated S cleavage

Catalysts with varying electronic structures provide unique characteristics, which affect electron donation and supply to adjacent

sulfur (S) species, thus regulating S redox processes with distinctive pathway selectivity. As depicted in Fig. 5a, b, the evolutions of Mn K-edge and Ni K-edge spectra are observed during the first cycle through ex-situ X-ray absorption spectroscopy (XAS), as confirmed by the positional shifts of the white line peaks in the XANES spectra. In the Mn K-edge (Fig. 5a) a low-intensity additional peak appears at 6537.8 eV, which is ascribed to the electron transition (ET) from the 1s orbital to the unoccupied 3d orbital, indicating the excitation of core-level electrons to the valence shell. In contrast, no evident peak is observed in the Ni K-edge pre-edge (Fig. 5b). Moreover, the Mn K-edge shifts by about 0.63 eV to a higher energy when the discharge voltage is set to 2.0 V (Fig. 5c), indicating that Mn₁ quickly transfers many electrons to S species. When discharged to 1.2 V and 0.8 V, the energy shifts decreased to 0.3 eV. This signifies that electrons are still being supplied to S species. In contrast, the Ni K-edge experiences no energy shifts in the high-voltage region, while there are increased energy shifts from 1.2 V to 0.8 V, implying that, due to the inferior ET capability of Ni₁, it can only donate electrons to S species in the low-voltage region.

During charging processes, the Mn K-edge progressively shifts to lower energy, reflecting a gradual reduction of Mn valence states. This suggests that Mn₁ gradually accepts electrons from S species, promoting S cathode oxidation reactions (Fig. 5d). Conversely, the Ni K-edge initially shifts to higher energy before moving to lower energy as the voltage changes from 1.2 V to 0.8 V, demonstrating that Ni₁ accepts electrons at 0.8 V, a process unfavorable for the oxidation process during charging. Accordingly, the shift of the near-edge absorption energy successfully reveals that the various SACs have different abilities to donate and accept electrons, suggesting their diverse catalytic abilities. As illustrated in Fig. 5e, Mn typically exhibits multiple valences, including Mn¹⁺, Mn²⁺, Mn³⁺, and Mn⁴⁺, consistent with the redox potentials of S and its intermediates. The XANES spectra reveal that the valence state of Mn₁ in S@Mn₁-PNC is close to +2 (Supplementary Information, Fig. S25), indicating that Mn₁ can donate electrons to adjacent S species to facilitate the conversion to $Na_2S_2$ and $Na_2S$, and accept electrons from $Na_2S$ to enhance reversibility. Further, DFT calculations were conducted to evaluate the

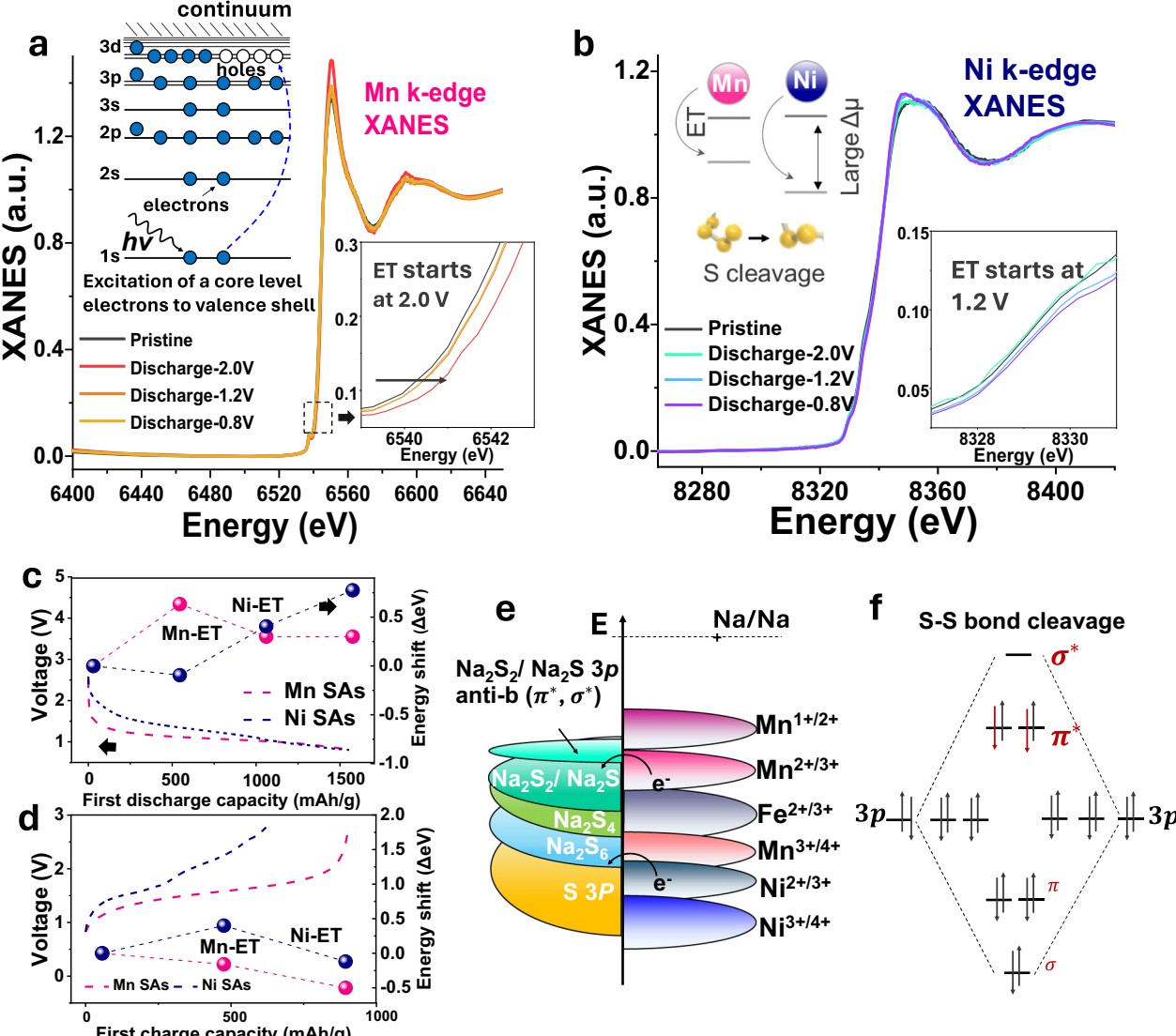

**Fig. 5 | Cleavage mechanism with fast kinetics of electron transfer from sulfur molecules. a** The Mn K-edge from ex-situ XANES spectra of the S@Mn₁-PNC sample during the discharge process, with the insets showing enlargements of the near-edge absorption energy shifts. **b** The Ni K-edge from ex-situ XANES spectra of the S@Ni₁-PNC sample during the discharge process, with the insets showing enlargements of the near-edge absorption energy shifts. **c** The energy shifts of S@Mn₁-PNC and S@Ni₁-PNC during the discharging process with their discharge curves. **d** The energy shifts of S@Mn₁-PNC and S@Ni₁-PNC during the charging process with their charge curves. **e** Schematic illustration of electron transfer caused by change in the electronic structure, where the voltage is determined by the energy gap. **f** Diagram of S-S orbital energy levels.

absorption energies of both $Mn_1$ and $Ni_1$ sites for $S_8$, NaPSs, and short-chain $Na_2S_2$ and $Na_2S$. The optimized absorption configurations are shown in Figure S26 and S27 (Supplementary information). As displayed in Figs. S26, S27, the ideal modes consisting of single atom Mn and Ni coordinated with 4 nitrogen atoms are applied in modeling the carbon matrix to calculate the absorption energy of various NaPSs. The energy absorption formula is defined as: $E(ad) = E(ad/surf) - E(surf) - E(ad)$, where E(ad/surf), E(surf), and E(ad) represent the total energies of the adsorbates binding to the surface, cleaning surface, and free adsorbate in gas phase, respectively. Thus, the absorption map in Figure S28 indicates $Mn_1$ sites exhibit stronger absorption abilities for NaPSs than $Ni_1$ sites, suggesting the S conversion reaction on $Mn_1$ sites is kinetically faster than that on $Ni_1$. This is consistent with the speculation from ex-situ XANES spectra. To better understand the product-selectivity of $Mn_1$, the correlations between $d$-band theory on five single-atom metals and their corresponding absorption energies are analyzed in Fig. S29. Overall, $Mn_1$ sites possess the strongest absorption abilities for $S_8$ and $Na_2S$ among different SA sites, suggesting that $Mn_1$ sites can effectively catalyze $S_8$ molecule cleavage and potentially produce $Na_2S$ as the primary reduction product. Moreover, the density of states (DOS) exhibits that the $d$-band states of $Mn_1$ sites are closer to the Fermi level than these of $Ni_1$ sites (Supplementary information, Fig. S30), demonstrating that the antibonding states of $Mn_1$ are less filled than that of $Ni_1$. Accordingly, the relationship between adsorption and the $d$-band center is negatively correlated (closer to Fermi level), in line with the conclusion of $d$-band theory. $Mn_1$ sites with the lowest $d$-band center, therefore, possess an increased likelihood of electrons filling the antibonding orbital, which facilitates S molecule cleavage and enhances the S redox kinetics, thus promoting product selectivity toward SC NaPSs (Fig. 5f)[56,57].

## Visualization of charge transfer assisted Na ion diffusion

ET has a positive effect on sodium ion diffusion, owing to the rearrangement of Na ions facilitated by the ET (Fig. 6a) according to Marcus's theory[58–60]. To verify this, in-situ transmission electron microscopy (in-situ TEM) was performed to visually investigate the sodium ion diffusion in the initial sodiation/desodiation processes (Fig. 6b, Supplementary Movie 1 and Supplementary Movie 2). As illustrated in Fig. 6c, the strength profile of $S@Mn_1$-PNC gradually increases during the discharging process, which means that the inside of the nanosphere is the first to react with the sodium ions. This is because ET causes an increase in the electrostatic potential, which enables quick adsorption of sodium ions. During the charging process, the strong ET capability of $Mn_1$ enables it to accept electrons from S species, which facilitates the desodiation process (Fig. 6d). When $Mn_1$ serves as a catalyst, S tends to be selectively converted to SC NaPSs, which might help mitigate volume expansion during S transformation processes by avoiding the formation of soluble NaPSs. Moreover, the ET effect of $Mn_1$ is revealed by monitoring the relative energy of sodium-ion diffusion on different matrixes (Supplementary information, Fig. S31). In contrast to the PNC matrix, the PNC matrix with anchored $Mn_1$ can significantly reduce the diffusion barriers of Na ions

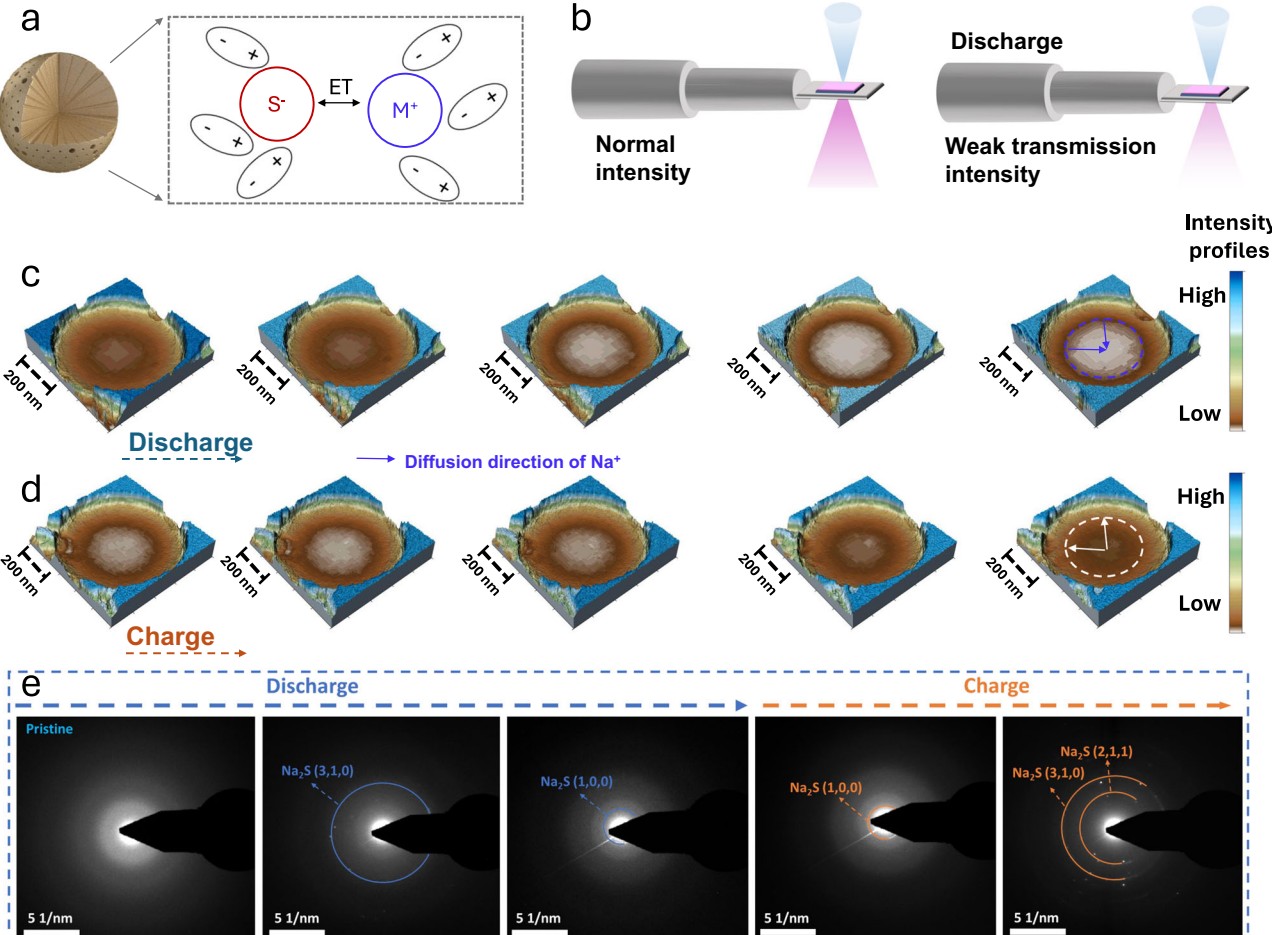

**Fig. 6 | Sodium ion diffusion visualization. a** Schematic diagram of the rearrangement of Na ions facilitated by ET. **b** Schematic diagram showing the view of an in-situ transmission electron microscope (TEM). **c** Sequential in-situ TEM images of S@Mn$_1$-PNC cathode during the initial discharge process. **d** Sequential in-situ TEM images of S@Mn$_1$-PNC cathode during the initial charge process. **e** In-situ SAED patterns of S@Mn$_1$-PNC cathode during the initial discharge and charge processes.

and facilitate redox reactions. Furthermore, the in-situ selected area electron diffraction (SAED) patterns reveal the phase conversion processes of S during the sodiation/de-sodiation processes (Fig. 6e). As the discharging processes continue, a new diffraction ring with a discernible reflection of $Na_2S$ emerges. More $Na_2S$ diffraction rings appear as sodiation increases, which is consistent with the in-situ XRD and ex-situ XAS results. After fully charging to 2.8 V, no other NaPS intermediates are observed besides $Na_2S$, indicating that $Na_2S$ consistently exists within the structure of S@Mn$_1$-PNC nanospheres. This resembles the lithiation processes in Li-S batteries, where only S and $Li_2S$ were detected, with no signs of intermediate products discovered by in-situ TEM[58,59].

## Discussion

The unique electronic structures of single-atom catalysts (SACs) showcase a variety of electron capture and donation capabilities, leading to different product-selectivity towards sodium polysulfide (NaPS) intermediates during the charge and discharge processes. This, in turn, results in diverse electrochemical performances in room temperature (RT) sodium-sulfur (Na-S) batteries. ML studies revealed the relationship between product selectivity and diverse single atom catalysts (SACs), thus predicting potential SACs for RT Na-S batteries. Guided by ML and DFT calculations, we investigated the performances of several single atom metal catalysts ($Mn_1$, $Fe_1$, $Co_1$, $Sn_1$, $Cu_1$, $Ni_1$) in cathodes for RT Na-S batteries. Among them, the single atom $Mn_1$ based cathode exhibited the best performance. Combining in-situ synchrotron powder XRD (PXRD) and ex-situ XAS, we successfully confirmed that $Mn_1$ single atoms possess unique ET capability, leading to high product selectivity towards short-chain NaPSs. Additionally, in-situ TEM and TOF-SIMS revealed that the stable structure of sulfur-loaded $Mn_1$-PNC effectively mitigates volume changes during the sulfur conversion reaction. This work could open a new avenue to the discovery of highly efficient single atom catalysts for diversified applications.

## Methods

### Synthesis methods

Fabrication of $Mn_1$-PNC and $M_1$-PNC (M = Fe, Ni, Co, Cu, Sn), where subscripted 1 stands for single atom. In a typical fabrication, 4 mg (0.01135 mmol) manganese acetylacetonate ($C_{10}H_{14}MnO_4$) and 0.5 g dopamine hydrochloride ($C_8H_{12}ClNO_2$) were dissolved into 4 mL ethanol and 6 mL deionized water to form a mixed solution, which was stirred for 1 h. This solution was labeled solution A. In addition, 36 mL ethanol and 94 mL deionized water were mixed at ambient temperature. Subsequently, 2 mL $NH_3 \cdot H_2O$ (28%) was dropped into the liquid mixture and kept under stirring for 30 min, with the product labeled as solution B. Solution A was poured into solution B with stirring at 400 rpm. After 10 h of stirring, the in-situ Mn precursor @PDA, where PDA is polydopamine, was collected after centrifugation and washed with deionized water three times. Immediately, the collected sample was dried in an oven at 60 °C overnight. Eventually, the $Mn_1$-PNC nanosphere was obtained via carbonization at 300 °C for 2 h and then increased to 800 °C for 2 h under $N_2$ atmosphere. The other single atomic samples were produced by using the same mole mass and same approach, except that different acetylacetonates were added. In the case of the $Fe_1$-PNC, the temperature of carbonization needed to be changed to 600 °C.

Synthesis of S@Mn$_1$-PNC and other S@M$_1$-PNC. The $Mn_1$-PNC was ground in an agate mortar with S powder with a mass ratio of 1:1.5. Then, the mixture was sealed in an ampoule, inserted into a quartz tube, and heated at 155 °C for 12 h. The temperature was then increased to 300 °C for 2 h at the heating rate of 5 °C min$^{-1}$. The obtained sample was denoted as S@Mn$_1$-PNC. Other S loaded single atom samples were obtained via the same procedures, the products of which were denoted as S@Fe$_1$-PNC, S@Co$_1$-PNC, S@Sn$_1$-PNC, S@Cu$_1$-PNC, and S@Ni$_1$-PNC, respectively.

### Electrode preparations and electrochemical performance measurements

First, the cathode slurry was produced by mixing the active materials, Super P, and carboxymethyl cellulose (CMC) binder in the ratio of 7:1.5:1.5 in distilled water, after which, the as-prepared slurry was coated on Cu foil by a clean blank and then loaded into a vacuum oven for 10 h at 50 °C. The electrodes were then punched with diameters of 0.95 cm and an average active material loading of 2.6 mg cm$^{-2}$. The cathodes for the RT Na-S battery cells were assembled in an argon-filled glove box with both $O_2$ and water less than 0.1 ppm. Na foil was employed as the negative electrode (reference electrode and counter electrode), and glass fiber (Whatman GF/Ft) was used as the separator. The electrolyte consisted of 1 M $NaClO_4$ in ethylene carbonate/propylene carbonate (EC/PC) with a volume ratio of 1:1, and 5 wt% fluoroethylene carbonate (FEC) as additive. The coin cells were investigated on NEWARE and Biologic VMP-3 electrochemical workstations within the voltage window of 0.8–2.8 V (vs. Na/Na$^+$). The NEWARE battery testers are placed at a lab with temperature set at 25 °C via air conditioner.

### Physical characterizations

X-ray diffraction (XRD) was conducted using Cu Kα radiation within the range of 10°–70° (GBC MMA diffractometer, $\lambda = 1.5406$ Å). The morphology was investigated with a field emission scanning electron microscope (FESEM, JEOL JSM-7500FA) equipped with energy-dispersive X-ray spectroscopy (EDS). In addition, a 200 kV scanning transmission electron microscope (STEM, JEM-ARM 200F) was used with a double aberration-corrector to acquire selected area electron diffraction (SAED) patterns with an image-forming lens system. The annular bright-field (ABF)-STEM images were collected with a STEM-ABF detector, and the angular range of collected electrons for high-angle annular dark-field (HAADF) images was about 70–250 mrad. The EDS mapping was processed by NSS software. TGA was conducted on a Mettler Toledo TGA/SDTA851 analyzer to measure the thermal decomposition behavior of samples in the temperature range from 50 °C to 900 °C with a heating rate of 5 °C min$^{-1}$. Time of flight secondary ion mass spectrometry (TOF-SIMS) measurements were carried out in negative mode for S and positive mode for Na based on the relative sensitivity factors (RSF) value. A 30 keV, 3 nA Ga$^+$ ion beam was utilized to sputter the cycled S@Mn$_1$-PNC electrode and produce the secondary ions. The analysis of TOF-SIMS data was performed using TOF-SIMS Explorer (Version: 1.3.1.0).

### Sample preparation for Atom probe tomography (APT) measurement

To prepare samples for the atom-probe experiments, the S@Mn$_1$-PNC nanospheres were cleaned and dried on a Si substrate (sample stage) and sputter coated with Cr in a sputter coater. After coating, specimens were transferred into a focus ion beam (FIB) system, where a strip of Pt was deposited on the top and surrounding area of the powder samples by using electron-assisted chemical vapor deposition. Afterwards, a microscope or SEM (scanning electron microscope) was performed to check the surface of the samples and mark the characterization positions. An FEI Quanta 200 3D FIB instrument was utilized to extract the samples[60]. This process involves cleaning the surrounding material to create a cantilevered area, followed by using a nano-hand to extract and cut it into multiple block samples from the prefabricated silicon base. During sample preparation, careful attention was paid to avoid damage from the Ga$^+$ focused ion beam, with ion energies kept below 10 keV after the lift out[61]. In addition, the sample's surface tends to accumulated impurities and residue during etching, thus necessitating thorough cleaning to ensure a smooth and high-quality surface. The APT measurements were conducted with a local electrode atom probe (LEAP 5000XR) under an ultraviolet (UV) laser pulsing at laser energy

of 200 pJ, a pulse repetition rate of 200 Da, and a target evaporation rate of 1.5% per pulse at 50 K. The reconstruction and quantitative analysis of the APT data were performed using CAMECA visualization and analysis software (AP Suite 6.1.3.42).

## In-situ characterizations

In-situ XRD data were collected at the Australian Synchrotron beamline at wavelengths of 0.6887 and 0.7223 Å. The X-ray absorption spectra were collected at the Japan Synchrotron Radiation Research Institute (JASRI) (1-1-1, Kouto, Sayo-cho, Sayo-gun, Hyogo 679-5198 Japan). The data were processed by Athena and Artemis software. In-situ transmission electron microscopy (TEM, FEIT Tecnai F20st) was conducted with a TEM-scanning tunneling microscope holder (Pico Femto FE02-ST) from Zeptools Co., Ltd.

## DFT calculations

All DFT calculations were performed on the Vienna ab-initio simulation package (VASP 5.4.4) based on spin-polarized density functional theory (DFT) methods. The generalized gradient approximation was used to estimate the exchange-correlation interaction in the scheme of the Perdew-Burke-Ernzerhof functional[62]. The interaction between core electrons and valence electrons was described using the projector augmented wave method[63]. The kinetic energy cut-off for the plane waves was set to 450 eV for the model calculations constructed with a $6 \times 6 \times 1$ graphene supercell. The convergence thresholds for energy and force on each atom during all structure optimizations were less than $10^{-5}$ eV and 0.02 eV/Å, respectively. To include the van der Waals force, the DFT-D3 method of Grimme was employed[64]. A vacuum distance of 15 Å along the *c*-axis was set to ensure sufficient vacuum and avoid interactions between two periods. For the calculations on the polysulfide decomposition energy barrier, the climbing image-nudged elastic band method was applied, and the force on each atom was kept below 0.05 eV Å$^{-1}$ [65–68].

## Machine learning (ML) methods

A method combining DFT calculations with ML by using the bond length of M-S and the adsorption energies of the metal with sulfur, $Na_2S$, and $Na_2S_4$ as indicators to predict advanced single atom catalysts for high-performance RT Na-S batteries. Here, a total of 123 adsorption energies of different $MN_4$ and $MC_1N_3$ sites were obtained by DFT calculations and 123 available DFT data (with the consideration of the stability of materials and thereby deleting the data with broken structures) were used for machine training and learning in five ML models (i.e., linear regression (LR), random forest regression (RFR), gradient boosted regression (GBR), support vector regression (SVR), and Kernel ridge regression (KRR) algorithms) coupling with the elemental information.

LR: linear regression; RFR: random forest regression; GBR: gradient boosted regression; SVR: support vector regression; KRR: Kernel ridge regression.

## Data availability

The data that support the findings of this work are available from the corresponding author upon reasonable request. Source data are provided with this paper.

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

## Acknowledgements

This research was supported by the Australian Research Council (ARC) (DE220101113, DP220103301, and DP210101389). The authors acknowledge the use of the facilities at the UOW electron Microscopy Centre (LE0882813 and LE0237478), Dr. Tania Silver for her critical reading, and Dr. Shaobo Li, who conducted the time-of-flight secondary-ion mass spectrometry.

## Author contributions

Y.-J.L., Y.F., L.Z., J.L. and S.W. conducted synthesis and characterization. X.L. contributed machine learning and density functional theory calculations. Y.-J.L. wrote the manuscript. W.-H.L. and Y.-X.W. supervised the project. H.Y. and D.M. performed X-ray absorption spectroscopy. Y.-J.L. and Q.-F.G. performed synchrotron X-ray diffraction measurements. D.S., T.R., Y.-X.W., H.-K.L., W.-H.L., S.-X.D., M.A., and G.W., helped draft the manuscript.

## Competing interests

The authors declare no competing interests.

## Additional information

[1]Institute for Superconducting & Electronic Materials, Australian Institute of Innovative Materials, University of Wollongong, Innovation Campus, Squires Way, North Wollongong, NSW 2500, Australia. [2]Centre for Clean Energy Technology, School of Mathematical and Physical Sciences, Faculty of Science, University of Technology Sydney, Sydney, NSW 2007, Australia. [3]School of Science and Technology, Kwansei Gakuin University, 2-1 Gakuen, Sanda, Hyogo 669-1337, Japan. [4]Australian Synchrotron 800 Blackburn Road, Clayton, VIC 3168, Australia. [5]Institute of Energy Materials Science, University of Shanghai for Science and Technology, Shanghai 200093, China. [6]Centre for Cooperative Research on Alternative Energies (CIC EnergiGUNE) Basque Research and Technology Alliance (BRTA) Alava Technology Park Albert Einstein 48, 01510 Vitoria-Gasteiz, Spain. [7]Inorganic Chemistry Department, University of the Basque Country UPV/EHU, P.O. Box. 644, 48080 Bilbao, Spain. [8]These authors contributed equally: Yao-Jie Lei, Xinxin Lu. ✉e-mail: weihongl@uow.edu.au; marmand@cicenergigune.com; yunxiaowang@usst.edu.cn; Guoxiu.Wang@uts.edu.au

