## [Peer Review File · Nature Communications]

Understanding the charge transfer effects of single atom for boosting the performance of Na-S batteriesREVIEWER COMMENTS

Reviewer #1 (Remarks to the Author):

Lei et al. reported a machine-learning-assisted approach to selecting single-atom catalysts for sodium-sulfur batteries in this paper. This approach focuses on catalysts that efficiently transfer electrons to the sulfur cathode while maintaining a low cleavage energy of the metal-sulfur bond. To validate the theoretical hypothesis, the authors synthesized six different types of single-atoms to demonstrate the generality of the approach. The optimized single-atoms demonstrated the capability to effectively catalyze the decomposition of the reaction product and thereby prevent the dissolution of polysulfides in Na-S batteries. The authors have achieved excellent performances for room-temperature Na-S batteries, highlighting the novelty and significance of this work. The entire manuscript is well written. The authors have done comprehensive computations, materials characterizations, battery testing, and in-depth analyses. Therefore, I recommend this manuscript to be accepted for publication in Nature Communications after minor revisions detailed below.

1. The detailed information on Ni single atoms (S@Ni¹-PNC) cathode materials should be provided in the manuscript.
2. Authors are advised to provide energy absorption maps and corresponding optimized structures for the active sites of Mn¹ and Ni¹ with respect to polysulfides.
3. In Figure S23, most d-band states of Mn are closer to the Fermi level than those of Ni, indicating the antibonding states of Mn are less filled. The density of states (DOS) of both Mn and Ni should be provided for further elucidation.
4. In the section "Visualization of charge transfer assisted Na ion diffusion", the authors showed that electron transfer has a positive effect on sodium ion diffusion. Therefore, it is necessary to provide the relative energy of sodium ion diffusion when Mn¹ is anchored on the matrix.
5. In Figure S14, the unit information is missing on the Y-axis, and the same issue is presented in Figure S20.

Reviewer #2 (Remarks to the Author):

This work reported an advanced material design strategy to develop single-atom catalysts (SACs) which enable an efficient capturing and donating electrons, influencing product selectivity in sodium polysulfide (NaPS) intermediates throughout charge and discharge processes in room temperature (RT) sodium-sulfur (Na-S) batteries. Through machine learning (ML) studies, correlations between product selectivity and different SACs were identified, enabling the prediction of potential SACs for RT Na-S batteries. High resolution HAADF results and atomic probe tomography shows the local structure and uniform distribution of SACs in sulfur cathode. The efficacy of selected single-atom metal catalysts (Mn¹, Fe¹, Co¹, Sn¹, Cu¹, Ni¹) in cathodes for RT Na-S batteries is evaluated by electrochemical measurement and their catalysis of charge transfer is full understood by in-situ synchrotron powder X-ray diffraction (PXRD) and ex-situ X-ray absorption spectroscopy (XAS). The manuscript is well written and organized. I would like to recommend its acceptance after necessary revisions.

What are the noteworthy results?

The design of material by combination of DFT and ML is quite advanced and the simulation results are inspiring for other catalysis research. Also the characterization of single atom

catalysis is novel and of high quality.

Will the work be of significance to the field and related fields? How does it compare to the established literature? If the work is not original, please provide relevant references.

The work will be very important to the development of Na-S batteries and also significant to the sulfur chemistry in rechargeable batteries. Both the simulation and characterization are beyond previous literatures.

Does the work support the conclusions and claims, or is additional evidence needed?

The conclusion is well supported by the data and some points for further clarification are provided below.

Are there any flaws in the data analysis, interpretation and conclusions? Do these prohibit publication or require revision?

There are some comments for further discussion but they do not prohibit publication.

Is the methodology sound? Does the work meet the expected standards in your field?

The methodologies are advanced and inspiring to surface chemistry in particular and to material design in general.

Is there enough detail provided in the methods for the work to be reproduced?

The methods were well described and some comments are provided below.

Here are comments for revision or discussion:

(1) The potential window is set as -1.91 eV to -0.61 eV and it is mentioned that "Based on voltage window conditions for..." But there is no reference to support this.

(2) In the description of equation 1, "HAD is the" should be "H (DA)".

(3) It is quite confusing to see that "the re-organization energy depends on the distance" but "it is often treated as a distance-independent". Dependent or independent? Please clarify this.

(4) In the sample preparation of atom probe tomography, more details are needed since it is not easy to prepare a cathode powder sample by the FIB.

(5) In figure 3 and b, the transition of sulfur to long chain and short chain species is very complicated and there is no clear border between them. It is good to say high plateau capacity and low plateau capacity. It will be tricky to identify long chain capacity from short chain capacity.

(6) In figure 4c and d, the text in scale bar is too small to read while the text in figure 4g/h is too big.

Reviewer #3 (Remarks to the Author):

This work comprehensively and insightfully explores the electronic structures of single-atom catalysts (SACs) in the context of room temperature (RT) sodium-sulfur (Na-S) batteries. The study effectively integrates machine learning (ML) techniques and density functional theory (DFT) calculations to establish a clear correlation between the electron capture and donation capabilities of various SACs and their respective product selectivity during charge and discharge processes. This manuscript can be considered for publications after the following questions are answered:

1. The reviewer is confused about the necessity of adoption of machine learning techniques

since the choice of different kinds of single-atom metals are limited. And the results could also be obtained through the DFT calculation. The reviewer suggests the author further explain the necessity of adopting the ML method.

2. The paper emphasized the application of ML in their research, but my feeling is that the authors only used some very rudimentary algorithms for fitting their results. ML did not show its power. I agree that the authors studied a fundamental problem. I also agree that ML + DFT is a quite exciting topic, but I cannot agree that this study used ML in a necessary and useful way. As a result, the ML aspect of this paper did not provide any insight for the readers.

3. Please clarify the size and structure of the data used for ML training. Is it only 69 simulation cases?

4. More details are needed to evaluate the authors' work on ML. It is currently not convincing at all.

5. Some axis labels are missing, for example Figure 4(a)(c)(d)

Responses to Reviewer #1:

Reviewer #1: Lei et al. reported a machine-learning-assisted approach to selecting single-atom catalysts for sodium-sulfur batteries in this paper. This approach focuses on catalysts that efficiently transfer electrons to the sulfur cathode while maintaining a low cleavage energy of the metal-sulfur bond. To validate the theoretical hypothesis, the authors synthesized six different types of single atoms to demonstrate the generality of the approach. The optimized single-atoms demonstrated the capability to effectively catalyze the decomposition of the reaction product and thereby prevent the dissolution of polysulfides in Na-S batteries. The authors have achieved excellent performances for room-temperature Na-S batteries, highlighting the novelty and significance of this work. The entire manuscript is well written. The authors have done comprehensive computations, materials characterizations, battery testing, and in-depth analyses. Therefore, I recommend this manuscript to be accepted for publication in Nature Communications after minor revisions detailed below.

Response: We appreciate the thorough comments and helpful suggestions from the referee. We have carefully addressed your feedback and made corresponding revisions to the manuscript, which are also outlined below in a point-by-point manner. All changes have been highlighted in the Revised Manuscript and Supplementary Information files.

1. The detailed information on Ni single atoms (S@Ni₁-PNC) cathode materials should be provided in the manuscript.

Response: Thanks for your valuable suggestion. To address this suggestion, we have added the SEM and high-resolution STEM images of S@Ni₁-PNC in Manuscript and Supplementary information, respectively.

Revision made:

(Manuscript, page 7)

For comparison, a sample with S loading on Ni₁-PNC (S@Ni₁-PNC) was analyzed by SEM, STEM and EDX mapping analysis (Supplementary Information, Figure S8 and S9), confirming that C, N, S, Ni are well-dispersed on the nanosphere with a diameter of ~500 nm.

(Supplementary information)

Figure S8. (a) SEM image of S@Ni₁-PNC nanospheres. (b) High-resolution TEM (HRTEM) image of S@Ni₁-PNC. (c) TEM image of S@Ni₁-PNC nanospheres in shell area.

Figure S9. Elemental mapping images of S@Ni₁-PNC nanospheres.

2. Authors are advised to provide energy absorption maps and corresponding optimized structures for the active sites of Mn₁ and Ni₁ with respect to polysulfides.

Response: Thanks for your valuable suggestion. The energy adsorption maps and corresponding optimized structures for both Mn₁ and Ni₁ with respect to polysulfides have been added in the revised Manuscript and revised Supplementary Information.

Revision made:

(Manuscript, page 15)

Further, DFT calculations were conducted to evaluate the absorption energies of both Mn₁ and Ni₁ sites for S₈, NaPSs, and short-chain Na₂S₂ and Na₂S. The optimized absorption configurations are shown in Figure S26 and S27 (Supplementary information). As displayed in Figure S26 and S27, the ideal modes consisting of single atom Mn and Ni coordinated with 4 nitrogen atoms are applied in modelling the carbon matrix to calculate the absorption energy of various NaPSs. The energy absorption formula is defined as: $E(\text{ad}) = E(\text{ad/surf}) - E(\text{surf}) - E(\text{ad})$, where E(ad/surf), E(surf), and E(ad) represent the total energies of the adsorbates binding to the surface, cleaning surface, and free adsorbate in gas phase, respectively. Thus, the absorption map in Figure S28 indicates Mn₁ sites exhibit stronger absorption abilities for NaPSs than Ni₁ sites, suggesting the S conversion reaction on Mn₁ sites is kinetically faster than that on Ni₁. This is consistent with the speculation from *ex-situ* XANES spectra.

(Supplementary information)

Figure S26. The adsorption configurations of Mn₁ to different polysulfides derived from DFT calculations.

Figure S27. The adsorption configurations of Ni₁ to different polysulfides derived from DFT calculations.

Figure S28. Adsorption energies of polysulfides on the active sites of Mn₁ and Ni₁.

3. In Figure S23, most d-band states of Mn are closer to the Fermi level than those of Ni, indicating the antibonding of Mn is less filled. The density of states (DOS) of both Mn and Ni should be provided for further elucidation.

Response: Thanks for your valuable comment. The density of states (DOS) of both Mn and Ni are provided in the revised Manuscript and Supplementary information. Additionally, Figure S23 is updated to Figure S30 after the correction.

Revision made:

(Manuscript, page 15)

To better understand the product-selectivity of Mn₁, the correlations between *d*-band theory on five single-atom metals and their corresponding absorption energies are analyzed in Figure S29. Overall, Mn₁ sites possess the strongest absorption abilities for S₈ and Na₂S among different SA sites, suggesting that Mn₁ sites can effectively catalyze S₈ molecule cleavage and potentially produce Na₂S as the primary reduction product. Moreover, the density of states (DOS) exhibits that the *d*-band states of Mn₁ sites are closer to the Fermi level than these of Ni₁ sites (Supplementary information, Figure S30), demonstrating that the antibonding states of Mn₁ are less filled than that of Ni₁. Accordingly, the relationship between adsorption and the *d*-band center is negatively correlated (closer to Fermi level), in line with the conclusion of *d*-band theory. Mn₁ sites with the lowest *d*-band center, therefore, possess an increased likelihood of electrons filling the antibonding orbital, which

facilitates S molecule cleavage and enhances the S redox kinetics, thus promoting product selectivity toward SC NaPSs (Figure 5f).

(Supplementary information)

Figure S30. The density of states (DOS) of both (a) S@Mn₁-PNC and (b) S@Ni₁-PNC surfaces.

4. In the section “Visualization of charge transfer assisted Na ion diffusion”, the authors showed that electron transfer has a positive effect on sodium ion diffusion. Therefore, it is necessary to provide the relative energy of sodium ion diffusion when Mn₁ is anchored on the matrix.

Response: We appreciate this great comment from the referee. The relative energy of sodium ion diffusion when Mn₁ is anchored on the matrix has been added in the manuscript and supplementary information.

Revision made:

(Manuscript, page 17)

Moreover, the electron transfer (ET) effect of Mn₁ is revealed by monitoring the relative energy of sodium-ion diffusion on different matrixes (Supplementary information, Figure S31). In contrast to the PNC matrix, the Mn₁-PNC matrix with anchored Mn₁ can significantly reduce the diffusion barriers of Na ions and facilitate redox reactions.

(Supplementary information)

Figure S31. (a) the relative energy of sodium-ion diffusion on PNC and Mn_1 -PNC with Mn_1 anchored on the matrix. (b) Schematic model of Na^+ diffusion path on PNC matrix. (c) Schematic model of Na^+ diffusion path on Mn_1 -PNC matrix.

5. In Figure S14, the unit information is missing on the Y-axis, and the same issue is presented in Figure S20.

Response: We appreciate the careful review. We have updated Figure S14 and Figure S20 in the Supplementary information. After correction, they are updated to Figure S18 and Figure 24.

Revision made:

(Supplementary information)

Figure S18. Charge-discharge curves at different current densities for S@Mn₁-PNC, S@Fe₁-PNC, S@Co₁-PNC, S@Sn₁-PNC, S@Ni₁-PNC, and S@Cu₁-PNC, respectively.

Figure S24. Time of flight-secondary ion mass spectroscopy (TOF-SIMS) images collected at various depths from the cathode surface of S@Mn₁-PNC illustrate the S and Na after 50 cycles.

Responses to Reviewer #2:

This work reported an advanced material design strategy to develop single-atom catalysts (SACs) which enable an efficient capturing and donating electrons, influencing product selectivity in sodium polysulfide (NaPS) intermediates throughout charge and discharge processes in room temperature (RT) sodium-sulfur (Na-S) batteries. Through machine learning (ML) studies, correlations between product selectivity and different SACs were identified, enabling the prediction of potential SACs for RT Na-S batteries. High resolution HAADF results and atomic probe tomography shows the local structure and uniform distribution of SACs in sulfur cathode. The efficacy of selected single-atom metal catalysts (Mn_1 , Fe_1 , Co_1 , Sn_1 , Cu_1 , Ni_1) in cathodes for RT Na-S batteries is evaluated by electrochemical measurement and their catalysis of charge transfer is fully understood by in-situ synchrotron powder X-ray diffraction (PXRD) and ex-situ X-ray absorption spectroscopy (XAS). The manuscript is well written and organized. I would like to recommend its acceptance after necessary revisions.

Response: We would like to thank the reviewer for the valuable comments and suggestions. We have revised the manuscript in accordance with the reviewers' comments and suggestions. Point-to-point responses are shown below.

(1) The potential window is set as -1.91 eV to -0.61eV and it is mentioned that "Based on voltage window conditions for..." But there is no reference to support this.

Response: We would like to thank the reviewer for carefully reading. First, the standard electrode potential for Na being ionized to form Na^+ is -2.71eV (vs RHE). The onset redox potentials of Na_2S_6 (long-chain polysulfides) is around 2.1V (versus Na/Na^+), which corresponds to -0.61 V (versus reversible hydrogen electrode (RHE)). Additionally, the cutoff discharge voltage is 0.8 V (versus Na/Na^+), which corresponds to -1.91 V (versus reversible hydrogen electrode (RHE)). This is why the potential window is set as -0.61eV to -1.91eV (vs. RHE). We have cited several representative works to support this. The corresponding references have been updated at the end of the description.

Revision made:

(Manuscript, page 5)

The onset redox potentials of Na_2S_6 is around 2.1V (versus Na/Na^+), which corresponds to -0.61 V (versus reversible hydrogen electrode (RHE)). Also, the cutoff discharge voltage is 0.8 V (versus Na/Na^+), which corresponds to -1.91 V (versus RHE). Based on these conditions for the battery test, it can be concluded that the cut-off chemical potential for all products obtained at the cathode is -0.61eV to -1.91 eV (versus RHE).^{37, 44, 45, 46,47}

References:

37. Hao H, Wang Y, Katyal N, Yang G, Dong H, Liu P, *et al.* Molybdenum Carbide Electrocatalyst In Situ Embedded in Porous Nitrogen-Rich Carbon Nanotubes Promotes Rapid Kinetics in Sodium-Metal-Sulfur Batteries. *Adv Mater* 2022, **34**(26): 2106572.
44. Zhang CY, Gong L, Zhang C, Cheng X, Balcells L, Zeng G, *et al.* Sodium-Sulfur Batteries with Unprecedented Capacity, Cycling Stability and Operation Temperature Range Enabled by a CoFe₂O₄ Catalytic Additive Under an External Magnetic Field. *Adv Funct Mater* 2023, **33**(48): 2305908.
45. Zhang E, Hu X, Meng L, Qiu M, Chen J, Liu Y, *et al.* Single-Atom Yttrium Engineering Janus Electrode for Rechargeable Na-S Batteries. *J Am Chem Soc* 2022, **144**(41): 18995-19007.
46. Li D, Gong B, Cheng X, Ling F, Zhao L, Yao Y, *et al.* An Efficient Strategy toward Multichambered Carbon Nanoboxes with Multiple Spatial Confinement for Advanced Sodium-Sulfur Batteries. *ACS Nano* 2021, **15**(12): 20607-20618.
47. Yang H, Ma X, Li Y, Zhou X, Chen L, Zhang Z, *et al.* Electrochemical redox kinetic behavior of S₈ and Na₂Sn (n = 2, 4, 6, 8) on vulcan XC-72R carbon in a flowing-electrolyte system. *J Power Sources* 2020, **478**: 229074.

(2) In the description of equation 1, "HAD is the" should be "H (DA)".

Response: We would like to thank the reviewer's comment. We have corrected "H_{AD}" to "H_(DA)" in this manuscript.

Revision made:

(Manuscript, page 5)

H_(DA) is the electronic coupling matrix element between donor and acceptor, ...

(3) It is quite confusing to see that " the re-organization energy depends on the distance " but " it is often treated as a distance-independent". Dependent or independent? Please clarify this.

Response: We thank for the reviewer's comment. We have corrected this in the manuscript.

Revision made:

(Manuscript, page 5)

The re-organization energy depends on the distance between the donor and acceptor (r_{DA}) and the solvent polarity (Eq. (2)).

(4) In the sample preparation of atom probe tomography, more details are needed since it is not easy to prepare a cathode powder sample by the FIB.

Response: Thank you for your suggestion. Preparing atomic probe tomography (APT) powder samples is more challenging than preparing bulk samples. The key difference is that, prior to FIB cutting, deposited Pt must cover not only the top of the powder sample but also the surrounding area of the powder sample. In contrast, for bulk samples, Pt only needs to be deposited on top of the bulk material. Detailed steps for preparing APT samples using FIB have been added in the revised supplementary information for the convenience of the readers.

Revision made:

(Supplementary information, Experimental section)

Sample preparation for Atom probe tomography (APT) measurement. To prepare samples for the atom-probe experiments, the S@Mn₁-PNC nanospheres were cleaned and dried on a Si substrate (sample stage) and sputter coated with Cr in a sputter coater. After coating, specimens were transferred into a focus ion beam (FIB) system, where a strip of Pt was deposited on the top and surrounding area of the powder samples by using electron-assisted chemical vapor deposition. Afterwards, a microscope or SEM (scanning electron microscope) was performed to check the surface of the samples and mark the characterization positions. An FEI Quanta 200 3D FIB instrument was utilized to extract the samples.¹ This process involves cleaning the surrounding material to create a cantilevered area, followed by using a nano-hand to extract and cut it into multiple block samples from the prefabricated silicon base. During sample preparation, careful attention was paid to avoid damage from the Ga⁺ focused ion beam, with ion energies kept below 10 keV after the lift out.² In addition, the sample's surface tends to accumulated impurities and residue during etching, thus necessitating thorough cleaning to ensure a smooth and high-quality surface.

References

1. K. Thompson, D. Lawrence, D. J. Larson, J. D. Olson, T. F. Kelly and B. Gorman, *Ultramicroscopy*, 2007, **107**, 131-139.
2. P. J. Felfer, T. Alam, S. P. Ringer and J. M. Cairney, *Microscopy Research and Technique*, 2012, **75**, 484-491.

(5) In figure 3 and b, the transition of sulfur to long chain and short chain species is very complicated and there is no clear boarder between them. It is good to say high plateau capacity and low plateau capacity. It will be tricky to identify long chain capacity from short chain capacity.

Response: Thank you very much for your comment. We agree that the transformation of sulfur from long-chain to short-chain species is very complex, and there is no clear boundary between them. Based on your comment, we have replaced the long-chain sulfur and short-chain sulfur in Figure 3a and 3b. Instead, the high plateau capacity and low plateau capacity are shown shown in Figure 3a and 3b. In addition, we have also made the corresponding revision in the revised manuscript.

Revision made:

(Manuscript, page 8, and 9)

Generally, the quasi-solid conversion reaction of S₈ cleavages can be divided into two regions in an ester-based electrolyte, including high-plateau capacity and low-plateau capacity. The capacities of these two plateaus are mainly contributed by long-chain (LC) NaPS (S₈→Na₂S_x→Na₂S₄, 2.8 V-1.25 V) and short-chain (SC) NaPS (Na₂S₄→Na₂S₂, Na₂S, 1.25 V-0.8 V) conversions, respectively. When Mn₁ acts as a catalyst for Na-S batteries, the discharge capacity primarily originates from low-plateau capacity, which indicates the capacity are mainly contributed by SC NaPS conversion, accounting for 88.6% of the total capacity (Figure 3c). In contrast, the high-plateau capacity region contributes 18.3% to the overall capacity. Such result indicates that the Mn₁ exhibits high pathway selectivity towards SC NaPS formation. Among the other five SACs, Fe₁ possesses similar pathway selectivity to Mn₁, in which the capacity contribution of the low-plateau region accounts for 81.7% of the total discharge capability. In the cases of the Co₁ and Sn₁, the SACs delivered 67% and 64% of low-plateau capacity, respectively. For Ni₁ and Cu₁, it is evident that the high-plateau capacity (S→Na₂S₆) region is dominant, suggesting that LC NaPS are main capacity contributor. Since the sluggish kinetics of SC conversion (low-plateau capacity) is the rate-determining step of the S redox reaction, it can be expected that Mn₁ would show the best performance, followed by Fe₁, Co₁, Sn₁, and finally Cu₁ and Ni₁. After 120 cycles at

a current density of 0.2 A g^{-1} , the $\text{S@Mn}_1\text{-PNC}$ cathode retains the highest capacity of 784.6 mAh g^{-1} amongst all the $\text{S@Mn}_1\text{-PNC}$ samples (Figure 3d), which also represents the highest capacity retention of 84%.

(Manuscript, page 10)

Overall, single atom catalysts exhibit distinct electrocatalytic activities towards the high-plateau and low-plateau conversion regions, showing different pathway selectivity. Meanwhile, the $\text{S@Mn}_1\text{-PNC}$ and $\text{S@Fe}_1\text{-PNC}$ cathodes demonstrate high reaction kinetics and excellent battery performance via effectively catalyzing the rate-determining step.

Figure 3. Product selectivity and electrochemical performance for $\text{S@Mn}_1\text{-PNC}$, $\text{S@Fe}_1\text{-PNC}$, $\text{S@Co}_1\text{-PNC}$, $\text{S@Sn}_1\text{-PNC}$, $\text{S@Ni}_1\text{-PNC}$, and $\text{S@Cu}_1\text{-PNC}$. (a) Initial discharge and (b) charge profiles. (c) The percentages of short-chain polysulfides in the initial discharges. (d) Cycling performances at current density of 0.2 mA g^{-1} . (e) Rate performances. (f) Long-term cycling performance of $\text{S@Mn}_1\text{-PNC}$ at the current density of 2 A g^{-1} .

(6) In figure 4c and d, the text in scale bar is too small to read while the text in figure 4g/h is too big.

Response: We thank the reviewer's careful reading. We increased the size in scale bar in figures 4c and 4d, while we decreased the size of the text in figures 4g and 4h in the manuscript. In addition, some axis labels are missing in Figure 4c and d. Accordingly, we added arrows from the surface to the interior in Figure 4c and Figure 4d, indicating that both Na and S are distributed throughout the bulk of the electrode.

Revision made:

Figure 4. Pathway selectivity and cycling stability mechanism. (a) *In-situ* synchrotron-based XRD patterns of S@Mn₁-PNC. (b) *Ex-situ* X-ray absorption spectra of S for S@Mn₁-PNC during the initial cycle. (c) and (d) 3D reconstructed images of TOF-SIMS depth profiles of Na and S after 10 cycles. (e) Normalized depth profiles of secondary ion fragments obtained from the S@Mn₁-PNC electrode after 50 cycles. (f) *k*³-weighted FT-EXAFS curves of S@Mn₁-PNC in R-space during the initial discharge and charge processes. WT plots of the Mn *k*-edge of S@Mn₁-PNC cathode during (g) discharge and (h) charge processes.

Responses to Reviewer #3:

This study provides a comprehensive and insightful exploration of the electronic structures of single-atom catalysts (SACs) in the context of room temperature (RT) sodium-sulfur (Na-S) batteries. It effectively integrates machine learning (ML) techniques and density functional theory (DFT) calculations to establish a clear correlation between the electron capture and donation capabilities of various SACs and their respective product selectivity during charge and discharge processes. This manuscript can be considered for publication after addressing the following questions:

We are very grateful for the in-depth and constructive comments and suggestions from the reviewer. We have revised the manuscript according to the reviewer's comments. All changes have been highlighted in the revised manuscript and supporting information files.

(1) The reviewer is confused about the necessity of adoption of machine learning techniques since the choice of different kinds of single-atom metals are limited. And the results could also be obtained through the DFT calculation. The reviewer suggests the author further explain the necessity of adopting the ML method.

Response: We thank the reviewer for this important question. Indeed, machine learning can give us a map that consists of energy differences and bond lengths between sulfur and metal. By combining this with knowledge of sodium-sulfur battery theory, we can quickly identify a specific area where a potential efficient catalyst may exist (as shown in Figure 1d). Nevertheless, this cannot be obtained by DFT calculations. Furthermore, Machine learning can integrate more scientific descriptors, making it simpler for researchers to pinpoint the most relevant descriptors quickly, filtering out the most effective materials. For instance, by utilizing the RandomForest method on 10 randomized data sets, we can swiftly identify the "Lms" 11 features, as illustrated in Figure S1. These features, determined by Pearson and Spearman methods, are closely linked to adsorption energy (Eds), aiding us in promptly identifying suitable sulfur cathode catalysts for sodium-sulfur batteries in our subsequent research alongside theory. In addition, while DFT simulations can evaluate the electrochemical performance of all metal monatomic structures, employing various machine learning algorithms for multidimensional correlation analysis and prediction of DFT-calculated datasets not only helps us recognize patterns of different physical and chemical properties' impact on adsorption or reaction energy but also allows for algorithm optimization with strong predictive capabilities through cross-correlation learning of numerous influencing descriptors. This optimized approach enables the prediction of potentially highly active metal atoms suitable for such reactions, a task that DFT simulations cannot achieve directly.

Revision made:

(Supplementary information)

Figure S1. Importance of used 11 features through the RandomForest method over the 10 randomized data, and Pearson and Spearman methods.

(2) The paper emphasized the application of ML in their research, but my feeling is that the authors only used some very rudimentary algorithms for fitting their results. ML did not show its power. I agree that the authors studied a fundamental problem. I also agree that ML + DFT is a quite exciting topic, but I cannot agree that this study used ML in a necessary and useful way. As a result, the ML aspect of this paper did not provide any insight for the readers.

Response: We appreciate the reviewer for bringing up this important point. Firstly, it is crucial to highlight that this work is centered on actively delving into the field of sodium-sulfur batteries. In the presenting work, we only have access to fewer than 20 individual atomic metals to select from the periodic table. This is the reason that the presented datasets are not as many as in other ML works. Understand this circumstance, we agree with the reviewer's concern that our current work may not fully exploit the power of machine learning for big data analysis. To avoid misunderstanding, we have made corresponding revisions in this article, including title, abstract, introduction and section 1. The revisions include two parts: increase the weight of experimental conclusions and optimize the number of ML dataset. The changes aim for readers to understand that our focus is primarily on the experimental exploration and identification of which single-atom's physical properties are crucial indicators for the selectivity of sulfur cathode products. However, we still believe that the machine learning in this article is an inspiring tool to build and screen the most suitable single atom catalysts for electron transfer and the lowest M-S bond energy with combination of our theoretical knowledge in sodium-sulfur batteries, to achieve the optimal discharge product (short-chain polysulfides). In particular, the experimental demonstration results and the ML prediction results are well consistent.

To increase accuracy, we extend the current 69 DFT simulation cases to 123 cases by adding a new structure M-N₃C into the presenting work. A quick note to emphasize the necessity of ML is that the ML can quickly identify the best scientific descriptors for evaluating the effects of various physical and chemical properties of the metal atoms in sodium sulfur batteries (as seen in Figure S1) and make predictions. This unique and important feature cannot be provided by DFT calculations.

Revision made:

(Title)

Understanding the charge transfer effects of single atoms for boosting the performance of Na-S batteries

(Manuscript, page 1)

The establishment of the synergistic interaction between the adsorption model and electronic transfer helps us achieve a high level of selectivity towards the desirable short-chain sodium polysulfides during the practical battery test.

((Manuscript, page 3)

In this study, we first construct the collaborative relationship between the absorption model and electron transfer. This method can help us rapidly screen out promising single atom catalysts that have a high level of selectivity towards the short-chain sodium polysulfides for sodium sulfur batteries.

(Manuscript, page 6)

Next, machine learning is performed as a sufficient way to identify the best scientific descriptors, which can assess how various physical and chemical properties of metal atoms affect adsorption or reaction energy. As shown in Figure S1, it is obviously that the bond length of metal and sulfur can influence the adsorption and reaction energy. According to the prediction in Figure 1e, a linear relationship between the adsorption energy E_{ads} and diverse SACs is obtained, which demonstrates that the prediction results of ML are like those of DFT calculations. Then, by establishing a linear relationship between the lengths of adsorption and metal-sulfur bonds (Figure 1d), combined with the theoretical knowledge discussion, we can rapidly screen potential SACs for Na-S batteries that should exist in a specific region (as shown as Figure 1d, optimized region), in which the SACs, Mn-N₄, Fe-N₄, Rh-N₄, Mg-N₄, Co-N₄ and Mg-C₁N₃ feature mild adsorption (more detailed machine learning procedures are in the Supplementary information, Figure S2, Figure S3, Figure S4 and Table S2, Table S3, Table S4, Table S5). With the assistance of prediction, we selected six representative SACs, Mn₁, Fe₁, Co₁, Sn₁, Cu₁, and Ni₁, which were used to conduct further experimental validation in Na-S batteries.

(Supplementary information, experimental section)

It is known that most correlations between the descriptors and targets were probably nonlinear in electrocatalytic and photocatalytic reactions. Herein, four types of nonlinear ML algorithms except linear regression (LR) algorithm were used to predict the adsorption activity in this work, including tree ensemble methods (RFR and GBR), and kernel methods (SVR and KRR) algorithms. The model performances of LR, RFR, GBR, SVR and KRR were compared by the root-mean-square error (RMSE) via k-fold cross validation (20-fold used in this case). The DFT dataset was randomly split into training and test sets, where 10% of the dataset is divided as a test set, and the other of the dataset becomes a train set. For a better machine learning (ML) performance, the standardization of target values was applied to our input dataset. In addition, after a rough screening to features according to the degree of feature importance, the 11 main features (Table S1) are chosen as descriptors. Package *sklearn* was employed to implement the data processing and import the ML algorithms.

LR: linear regression; RFR: random forest regression; GBR: gradient boosted regression; SVR: support vector regression; KRR: Kernel ridge regression

Table S1. List of DFT-calculated and elemental features used as descriptors.

Features	
Period	Period number of metal element

RWIGS	Bulk wigner-seitz radius of metal element
Rm	Atomic radius
Nve	Valence electron number of metal element
Am	Electron affinity
Mm	Atomic mass of metal element
Xm	Electronegativity of metal element
Dm	Density of metal element
Ms	Atomic mass of adsorption species
Lms	Length of metal-sufur bond
D_Lmn	Average change of metal-nitrogen bond length

(Supplementary information)

Figures and Tables

Figure S1. Importance of used 11 features through the RandomForest method over the 10 randomized data, and Pearson and Spearman methods.

Table S2. Values of important 11 features through the RandomForest method over the 10 randomized data, and Pearson and Spearman methods.

	Pearson	Spearman	RandomForest
Period	0.077	0.15	0.005
Group	0.369	0.394	0.142
RWIGS	0.074	0.111	0.024
Nve	0.389	0.405	0.089

Am	0.22	0.293	0.019
Mm	0.118	0.278	0.016
Xm	0.052	0.137	0.015
Dm	0.093	0.163	0.014
Ms	0.577	0.58	0.203
Lms	0.502	0.639	0.412
D Lmn	0.429	0.6	0.061

A method combining density functional theory (DFT) calculations with machine learning (ML) by using the bond length of M-S and the adsorption energies of the metal with sulfur, Na₂S, and Na₂S₄ as indicators to predict advanced single atom catalysts for high-performance RT Na-S batteries. Here, a total of 123 adsorption energies of different MN₄ and MC₁N₃ sites were obtained by DFT calculations and 123 available DFT data (with the consideration of the stability of materials and thereby deleting the data with broken structures) were used for machine training and learning in five ML models (i.e. linear regression (LR), random forest regression (RFR), gradient boosted regression (GBR), support vector regression (SVR), and Kernel ridge regression (KRR) algorithms) coupling with the elemental information. The relationships between the used descriptors and adsorption activity were firstly analyzed through the Pearson correlation coefficient (Pearson), the Spearman correlation coefficient (Spearman) and the RandomForest feature importance (RandomForest) methods, as shown in Figure S1 and Table S2. The rankings of these descriptors by Pearson are basically consistent with that by Spearman, except for the metal-sulfur bond (Lms), whereas RandomForest method displays obviously distinction although the top two rankings are same with Spearman method. The rankings indicate that adsorption energies to polysulfides and sodium sulfide exhibits low correlations with most descriptors except the change of metal-nitrogen bond length (D Lmn), Lms and valence electron number of metal element (Nve) and adsorbed species (Ms). After a rough screening of features according to the degree of feature importance, 11 main features (Figure S1) were chosen as descriptors. The *sklearn* package was employed to implement the data processing and import the ML algorithms.

Figure S2. Comparison of the adsorption energy (E_{ds}) from the DFT calculations and the full-fit results using the various ML algorithms: (a) RFR and (b) LR, respectively. R^2 : coefficient of determination; RMSE: root mean square error.

Figure S3. Predicted vs. DFT-calculated E_{ds} of single-atom catalyst (SAC) materials by using SVR (a), and KRR (b) algorithms, with 80% training (blue dot) and 20% testing (red dot) data.

Table S3. Feature values for metal elements in ML modelling, including atomic number of metal element, period number of metal element, the group of metal element, bulk wigner-seitz radius of metal element, valence electron number of metal element, atomic mass of metal element, electronegativity of metal element, density

of metal element.

Metal	Period	Group	RWIGS	Nve	Am	Mm	Xm	Dm	Ms	Lms	D Lmn
MgN ₄	3	2	2.88	2	-40	24.03	1.31	1.74	256.48	2.773	-0.03
MgN ₄	3	2	2.88	2	-40	24.03	1.31	1.74	110.12	2.402	-0.126
MgN ₄	3	2	2.88	2	-40	24.03	1.31	1.74	78.06	2.62	-0.112
MgCN ₃	3	2	2.88	2	-40	24.03	1.31	1.74	256.48	2.705	-0.046
MgCN ₃	3	2	2.88	2	-40	24.03	1.31	1.74	110.12	2.511	-0.105
MgCN ₃	3	2	2.88	2	-40	24.03	1.31	1.74	78.06	2.414	-0.159
AlN ₄	3	13	2.65	3	41.76	26.98	1.61	2.7	256.48	2.621	-0.017
AlN ₄	3	13	2.65	3	41.76	26.98	1.61	2.7	110.12	2.24	-0.084
AlN ₄	3	13	2.65	3	41.76	26.98	1.61	2.7	78.06	2.322	-0.058
AlCN ₃	3	13	2.65	3	41.76	26.98	1.61	2.7	256.48	2.653	-0.011
AlCN ₃	3	13	2.65	3	41.76	26.98	1.61	2.7	110.12	2.343	-0.055
AlCN ₃	3	13	2.65	3	41.76	26.98	1.61	2.7	78.06	2.261	-0.084
SiN ₄	3	14	2.48	4	134.06	28.08	1.9	2.33	256.48	2.184	-0.071
SiN ₄	3	14	2.48	4	134.06	28.08	1.9	2.33	110.12	2.128	-0.078
SiN ₄	3	14	2.48	4	134.06	28.08	1.9	2.33	78.06	2.213	-0.057
SiCN ₃	3	14	2.48	4	134.06	28.08	1.9	2.33	256.48	2.657	-0.008
SiCN ₃	3	14	2.48	4	134.06	28.08	1.9	2.33	110.12	2.228	-0.052
SiCN ₃	3	14	2.48	4	134.06	28.08	1.9	2.33	78.06	2.148	-0.073
TiN ₄	4	4	2.5	4	7.28	47.87	1.54	4.51	256.48	2.43	-0.032
TiN ₄	4	4	2.5	4	7.28	47.87	1.54	4.51	110.12	2.261	0.028
TiN ₄	4	4	2.5	4	7.28	47.87	1.54	4.51	78.06	2.414	0.017
TiCN ₃	4	4	2.5	4	7.28	47.87	1.54	4.51	256.48	2.355	-0.01

TiCN ₃	4	4	2.5	4	7.28	47.87	1.54	4.51	110.12	2.412	-0.035
TiCN ₃	4	4	2.5	4	7.28	47.87	1.54	4.51	78.06	2.27	-0.018
MnN ₄	4	7	2.5	7	-50	54.94	1.55	7.21	256.48	2.656	-0.008
MnN ₄	4	7	2.5	7	-50	54.94	1.55	7.21	110.12	2.3	-0.042
MnN ₄	4	7	2.5	7	-50	54.94	1.55	7.21	78.06	2.388	-0.023
MnCN ₃	4	7	2.5	7	-50	54.94	1.55	7.21	256.48	2.556	-0.013
MnCN ₃	4	7	2.5	7	-50	54.94	1.55	7.21	110.12	2.28	-0.018
MnCN ₃	4	7	2.5	7	-50	54.94	1.55	7.21	78.06	2.218	-0.032
FeN ₄	4	8	2.46	8	14.78	55.85	1.83	7.87	256.48	2.641	-0.011
FeN ₄	4	8	2.46	8	14.78	55.85	1.83	7.87	110.12	2.213	-0.013
FeN ₄	4	8	2.46	8	14.78	55.85	1.83	7.87	78.06	2.182	-0.013
FeCN ₃	4	8	2.46	8	14.78	55.85	1.83	7.87	256.48	2.338	-0.014
FeCN ₃	4	8	2.46	8	14.78	55.85	1.83	7.87	110.12	2.181	-0.013
FeCN ₃	4	8	2.46	8	14.78	55.85	1.83	7.87	78.06	2.179	-0.024
CoN ₄	4	9	2.46	9	63.89	58.93	1.88	8.86	256.48	2.447	-0.025
CoN ₄	4	9	2.46	9	63.89	58.93	1.88	8.86	110.12	2.258	-0.02
CoN ₄	4	9	2.46	9	63.89	58.93	1.88	8.86	78.06	2.244	-0.022
CoCN ₃	4	9	2.46	9	63.89	58.93	1.88	8.86	256.48	2.26	-0.033
CoCN ₃	4	9	2.46	9	63.89	58.93	1.88	8.86	110.12	2.235	-0.025
CoCN ₃	4	9	2.46	9	63.89	58.93	1.88	8.86	78.06	2.259	-0.023
NiN ₄	4	10	2.43	10	111.65	58.69	1.91	8.9	256.48	3.288	-0.002
NiN ₄	4	10	2.43	10	111.65	58.69	1.91	8.9	110.12	2.502	-0.012
NiN ₄	4	10	2.43	10	111.65	58.69	1.91	8.9	78.06	3.86	0
NiCN ₃	4	10	2.43	10	111.65	58.69	1.91	8.9	256.48	3.22	-0.002
NiCN ₃	4	10	2.43	10	111.65	58.69	1.91	8.9	110.12	3.935	-0.002
NiCN ₃	4	10	2.43	10	111.65	58.69	1.91	8.9	78.06	2.439	-0.019
CuN ₄	4	11	2.2	11	119.23	63.55	1.9	8.96	256.48	3.276	-0.001
CuN ₄	4	11	2.2	11	119.23	63.55	1.9	8.96	110.12	2.502	-0.047
CuN ₄	4	11	2.2	11	119.23	63.55	1.9	8.96	78.06	3.77	0.001
CuCN ₃	4	11	2.2	11	119.23	63.55	1.9	8.96	256.48	3.281	-0.001
CuCN ₃	4	11	2.2	11	119.23	63.55	1.9	8.96	110.12	2.615	-0.023
CuCN ₃	4	11	2.2	11	119.23	63.55	1.9	8.96	78.06	2.471	-0.034
ZnN ₄	4	12	2.4	12	-58	65.38	1.65	7.14	256.48	2.79	-0.019
ZnN ₄	4	12	2.4	12	-58	65.38	1.65	7.14	110.12	2.266	-0.151
ZnN ₄	4	12	2.4	12	-58	65.38	1.65	7.14	78.06	2.327	-0.104
ZnCN ₃	4	12	2.4	12	-58	65.38	1.65	7.14	256.48	3.417	-0.015
ZnCN ₃	4	12	2.4	12	-58	65.38	1.65	7.14	110.12	2.376	-0.104
ZnCN ₃	4	12	2.4	12	-58	65.38	1.65	7.14	78.06	2.278	-0.187
MoN ₄	5	6	2.75	6	72.1	95.96	2.16	10.2 8	256.48	2.2	-0.023

MoN₄	5	6	2.75	6	72.1	95.96	2.16	$\frac{10.2}{8}$	110.12	2.242	-0.027
MoN₄	5	6	2.75	6	72.1	95.96	2.16	$\frac{10.2}{8}$	78.06	2.375	-0.106
MoCN₃	5	6	2.75	6	72.1	95.96	2.16	$\frac{10.2}{8}$	256.48	2.284	-0.077
MoCN₃	5	6	2.75	6	72.1	95.96	2.16	$\frac{10.2}{8}$	110.12	2.359	-0.107
MoCN₃	5	6	2.75	6	72.1	95.96	2.16	$\frac{10.2}{8}$	78.06	2.245	-0.009
RuN₄	5	8	2.65	8	100.27	101.1	2.2	$\frac{12.4}{5}$	256.48	2.136	-0.037
RuN₄	5	8	2.65	8	100.27	101.1	2.2	$\frac{12.4}{5}$	110.12	2.294	-0.009
RuN₄	5	8	2.65	8	100.27	101.1	2.2	$\frac{12.4}{5}$	78.06	2.441	-0.072
RuCN₃	5	8	2.65	8	100.27	101.1	2.2	$\frac{12.4}{5}$	256.48	2.146	-0.049
RuCN₃	5	8	2.65	8	100.27	101.1	2.2	$\frac{12.4}{5}$	110.12	2.331	-0.065
RuCN₃	5	8	2.65	8	100.27	101.1	2.2	$\frac{12.4}{5}$	78.06	2.261	-0.032
RhN₄	5	9	2.65	9	100.27	102.9	2.28	$\frac{12.4}{1}$	256.48	3.078	-0.002
RhN₄	5	9	2.65	9	100.27	102.9	2.28	$\frac{12.4}{1}$	110.12	2.334	-0.018
RhN₄	5	9	2.65	9	100.27	102.9	2.28	$\frac{12.4}{1}$	78.06	4.118	-0.005
RhCN₃	5	9	2.65	9	100.27	102.9	2.28	$\frac{12.4}{1}$	256.48	2.304	-0.038
RhCN₃	5	9	2.65	9	100.27	102.9	2.28	$\frac{12.4}{1}$	110.12	2.319	-0.03
RhCN₃	5	9	2.65	9	100.27	102.9	2.28	$\frac{12.4}{1}$	78.06	2.341	-0.023
PdN₄	5	10	2.71	10	54.24	106.4	2.2	$\frac{12.0}{2}$	256.48	3.486	-0.003
PdN₄	5	10	2.71	10	54.24	106.4	2.2	$\frac{12.0}{2}$	110.12	2.728	-0.006
PdN₄	5	10	2.71	10	54.24	106.4	2.2	$\frac{12.0}{2}$	78.06	3.95	-0.002
PdCN₃	5	10	2.71	10	54.24	106.4	2.2	$\frac{12.0}{2}$	256.48	3.477	-0.003
PdCN₃	5	10	2.71	10	54.24	106.4	2.2	$\frac{12.0}{2}$	110.12	4.002	-0.004
PdCN₃	5	10	2.71	10	54.24	106.4	2.2	$\frac{12.0}{2}$	78.06	2.659	-0.01
AgN₄	5	11	2.84	11	125.86	107.9	1.93	$\frac{10.4}{9}$	256.48	3.443	-0.005

AgN ₄	5	11	2.84	11	125.86	107.9	1.93	$\frac{10.4}{9}$	78.06	3.63	-0.006
AgCN ₃	5	11	2.84	11	125.86	107.9	1.93	$\frac{10.4}{9}$	256.48	3.472	-0.003
AgCN ₃	5	11	2.84	11	125.86	107.9	1.93	$\frac{10.4}{9}$	110.12	3.154	-0.005
AgCN ₃	5	11	2.84	11	125.86	107.9	1.93	$\frac{10.4}{9}$	78.06	2.694	-0.037
SnN ₄	5	14	2.96	4	107.29	118.7	1.96	7.26	256.48	3.866	0.002
SnN ₄	5	14	2.96	4	107.29	118.7	1.96	7.26	110.12	2.378	0.156
SnN ₄	5	14	2.96	4	107.29	118.7	1.96	7.26	78.06	4.024	0.027
SnCN ₃	5	14	2.96	4	107.29	118.7	1.96	7.26	256.48	2.516	0.058
SnCN ₃	5	14	2.96	4	107.29	118.7	1.96	7.26	110.12	2.481	-0.035
SnCN ₃	5	14	2.96	4	107.29	118.7	1.96	7.26	78.06	5.535	-0.029
WN ₄	6	6	2.75	6	78.76	183.8	2.36	19.3	256.48	2.139	-0.055
WN ₄	6	6	2.75	6	78.76	183.8	2.36	19.3	110.12	2.236	-0.036
WN ₄	6	6	2.75	6	78.76	183.8	2.36	19.3	78.06	2.355	-0.128
WCN ₃	6	6	2.75	6	78.76	183.8	2.36	19.3	256.48	2.353	-0.083
WCN ₃	6	6	2.75	6	78.76	183.8	2.36	19.3	110.12	2.328	-0.142
WCN ₃	6	6	2.75	6	78.76	183.8	2.36	19.3	78.06	2.239	-0.032
IrN ₄	6	9	2.84	9	150.94	192.2	2.2	$\frac{22.5}{6}$	256.48	3.392	-0.002
IrN ₄	6	9	2.84	9	150.94	192.2	2.2	$\frac{22.5}{6}$	110.12	2.331	-0.024
IrN ₄	6	9	2.84	9	150.94	192.2	2.2	$\frac{22.5}{6}$	78.06	4.2	-0.007
IrCN ₃	6	9	2.84	9	150.94	192.2	2.2	$\frac{22.5}{6}$	256.48	2.259	-0.045
IrCN ₃	6	9	2.84	9	150.94	192.2	2.2	$\frac{22.5}{6}$	110.12	2.316	-0.041
IrCN ₃	6	9	2.84	9	150.94	192.2	2.2	$\frac{22.5}{6}$	78.06	2.337	-0.031
PtN ₄	6	10	2.75	10	205.04	195.1	2.28	$\frac{21.4}{5}$	256.48	3.584	-0.002
PtN ₄	6	10	2.75	10	205.04	195.1	2.28	$\frac{21.4}{5}$	110.12	2.69	-0.009
PtN ₄	6	10	2.75	10	205.04	195.1	2.28	$\frac{21.4}{5}$	78.06	3.999	-0.003
PtCN ₃	6	10	2.75	10	205.04	195.1	2.28	$\frac{21.4}{5}$	256.48	3.552	-0.003
PtCN ₃	6	10	2.75	10	205.04	195.1	2.28	$\frac{21.4}{5}$	110.12	4.02	-0.005
PtCN ₃	6	10	2.75	10	205.04	195.1	2.28	$\frac{21.4}{5}$	78.06	2.621	-0.012
AuN ₄	6	11	2.84	11	222.75	197	2.54	$\frac{19.3}{2}$	256.48	3.542	-0.002

AuN ₄	6	11	2.84	11	222.75	197	2.54	$\frac{19.3}{2}$	110.12	3.69	-0.002
AuN ₄	6	11	2.84	11	222.75	197	2.54	$\frac{19.3}{2}$	78.06	3.063	-0.004
AuCN ₃	6	11	2.84	11	222.75	197	2.54	$\frac{19.3}{2}$	256.48	3.53	-0.003
AuCN ₃	6	11	2.84	11	222.75	197	2.54	$\frac{19.3}{2}$	110.12	3.664	-0.003
AuCN ₃	6	11	2.84	11	222.75	197	2.54	$\frac{19.3}{2}$	78.06	3.084	-0.003
BiN ₄	6	15	3.09	5	90.92	209	2.02	9.78	256.48	3.746	0.003
BiN ₄	6	15	3.09	5	90.92	209	2.02	9.78	78.06	3.015	-0.08
BiCN ₃	6	15	3.09	5	90.92	209	2.02	9.78	256.48	3.721	-0.004
BiCN ₃	6	15	3.09	5	90.92	209	2.02	9.78	78.06	2.679	-0.146

Table S4. Eds from the DFT calculations and the full-fit results using the three ML algorithms, LR, RFR and GBR, respectively.

Metal	Eds	LR-pred	RFR-pred	GBR-pred	SVR-pred	KRR-pred
MgN ₄	-0.684	-0.732	-0.702	-0.714	-0.793	-0.997
MgN ₄	-1.6104	-2.075	-1.841	-1.689	-1.869	-2.033
MgN ₄	-1.9576	-2.153	-1.888	-1.894	-2.004	-2.149
MgCN ₃	-0.72	-0.820	-0.736	-0.713	-0.793	-1.028
MgCN ₃	-1.8096	-1.958	-1.811	-1.793	-1.869	-1.986
MgCN ₃	-1.8728	-2.409	-1.939	-1.899	-2.004	-2.244
AlN ₄	-0.7832	-0.801	-0.755	-0.713	-0.841	-1.006
AlN ₄	-2.2632	-2.010	-2.330	-2.289	-1.962	-2.021
AlN ₄	-2.624	-2.055	-2.453	-2.554	-2.080	-2.165
AlCN ₃	-0.7328	-0.768	-0.735	-0.707	-0.841	-0.993
AlCN ₃	-2.1344	-1.856	-2.294	-2.236	-1.962	-1.970
AlCN ₃	-2.344	-2.188	-2.358	-2.443	-2.080	-2.203
SiN ₄	-1.0936	-1.366	-1.370	-1.134	-0.749	-0.990
SiN ₄	-2.6448	-2.236	-2.689	-2.626	-1.445	-1.867
SiN ₄	-3.1856	-2.303	-2.915	-3.062	-1.513	-2.015
SiCN ₃	-0.68	-0.987	-0.675	-0.699	-0.749	-0.809
SiCN ₃	-2.44	-2.096	-2.579	-2.487	-1.444	-1.819
SiCN ₃	-2.7984	-2.390	-2.797	-2.809	-1.513	-2.045
TiN ₄	-2.1176	-1.980	-1.828	-2.084	-0.817	-1.477
TiN ₄	-3.2048	-2.555	-3.172	-3.238	-1.997	-2.331
TiN ₄	-3.556	-2.761	-3.263	-3.456	-2.128	-2.486
TiCN ₃	-1.4496	-1.890	-1.608	-1.535	-0.817	-1.480

TiCN ₃	-3.2888	-2.824	-3.124	-3.196	-1.997	-2.341
TiCN ₃	-3.4976	-2.951	-3.329	-3.395	-2.128	-2.554
MnN ₄	-1.014	-1.061	-0.764	-1.104	-0.751	-1.082
MnN ₄	-1.6904	-2.111	-1.882	-1.827	-1.818	-2.063
MnN ₄	-1.8976	-2.187	-1.949	-1.937	-1.955	-2.211
MnCN ₃	-0.7032	-1.103	-0.721	-0.730	-0.751	-1.114
MnCN ₃	-1.8312	-2.002	-1.890	-1.780	-1.818	-2.049
MnCN ₃	-2.172	-2.260	-2.122	-2.169	-1.955	-2.265
FeN ₄	-0.6936	-1.035	-0.705	-0.730	-0.794	-1.028
FeN ₄	-1.8984	-1.947	-1.892	-1.906	-1.975	-2.003
FeN ₄	-1.8032	-2.133	-1.852	-1.893	-2.106	-2.200
FeCN ₃	-0.728	-1.102	-0.950	-0.778	-0.794	-1.114
FeCN ₃	-1.9608	-1.952	-1.921	-1.904	-1.975	-2.012
FeCN ₃	-2.1496	-2.185	-2.048	-2.120	-2.106	-2.209
CoN ₄	-0.7152	-0.843	-0.713	-0.707	-0.750	-0.830
CoN ₄	-1.6528	-1.680	-1.673	-1.582	-1.830	-1.733
CoN ₄	-1.5288	-1.873	-1.620	-1.650	-1.944	-1.926
CoCN ₃	-0.7128	-0.914	-0.805	-0.725	-0.751	-0.887
CoCN ₃	-1.8872	-1.708	-1.842	-1.778	-1.830	-1.744
CoCN ₃	-1.7408	-1.875	-1.762	-1.799	-1.944	-1.923
NiN ₄	-0.568	-0.381	-0.572	-0.568	-0.671	-0.314
NiN ₄	-1.1848	-1.394	-1.213	-1.085	-1.516	-1.394
NiN ₄	-0.9792	-1.279	-1.013	-1.065	-1.603	-1.199
NiCN ₃	-0.5712	-0.393	-0.573	-0.568	-0.671	-0.333
NiCN ₃	-1.3448	-1.094	-1.268	-1.253	-1.515	-0.993
NiCN ₃	-1.1608	-1.619	-1.402	-1.424	-1.604	-1.604
CuN ₄	-0.588	-0.594	-0.592	-0.594	-0.641	-0.045
CuN ₄	-1.204	-1.773	-1.212	-1.204	-1.445	-1.151
CuN ₄	-1.12	-1.505	-1.055	-1.076	-1.528	-0.952
CuCN ₃	-0.5976	-0.593	-0.596	-0.594	-0.641	-0.044
CuCN ₃	-1.3448	-1.641	-1.277	-1.244	-1.445	-1.101
CuCN ₃	-1.2032	-1.899	-1.199	-1.231	-1.529	-1.337
ZnN ₄	-0.6288	-0.034	-0.635	-0.618	-0.729	-0.149
ZnN ₄	-1.7616	-1.573	-1.826	-1.761	-1.763	-1.256
ZnN ₄	-1.9504	-1.523	-1.896	-1.910	-1.899	-1.389
ZnCN ₃	-0.6488	0.095	-0.630	-0.663	-0.729	0.026
ZnCN ₃	-1.6904	-1.333	-1.797	-1.700	-1.763	-1.188
ZnCN ₃	-1.7992	-1.921	-1.863	-1.834	-1.899	-1.470
MoN ₄	-1.4016	-1.753	-1.556	-1.433	-0.706	-1.919

MoN ₄	-3.0064	-2.592	-3.057	-3.130	-1.793	-2.767
MoN ₄	-3.3312	-3.120	-3.309	-3.306	-1.905	-2.982
MoCN ₃	-1.9824	-1.992	-1.979	-1.930	-0.706	-1.940
MoCN ₃	-3.1168	-2.947	-3.135	-3.163	-1.793	-2.800
MoCN ₃	-3.432	-2.688	-3.306	-3.311	-1.905	-2.939
RuN ₄	-1.2136	-1.506	-1.181	-1.227	-0.634	-1.598
RuN ₄	-2.032	-2.174	-2.119	-2.069	-1.593	-2.388
RuN ₄	-2.484	-2.625	-2.070	-2.537	-1.692	-2.587
RuCN ₃	-1.4832	-1.561	-1.328	-1.397	-0.634	-1.605
RuCN ₃	-2.5888	-2.430	-2.298	-2.497	-1.593	-2.424
RuCN ₃	-2.8	-2.469	-2.593	-2.626	-1.692	-2.604
RhN ₄	-0.5304	-0.960	-0.539	-0.570	-0.627	-1.187
RhN ₄	-1.3616	-1.993	-1.531	-1.576	-1.587	-2.260
RhN ₄	-1.6704	-1.799	-1.537	-1.645	-1.685	-1.948
RhCN ₃	-0.6616	-1.265	-0.777	-0.716	-0.627	-1.429
RhCN ₃	-1.9968	-2.052	-1.958	-1.960	-1.587	-2.274
RhCN ₃	-2.1048	-2.197	-1.929	-1.873	-1.686	-2.450
PdN ₄	-0.5568	-0.462	-0.561	-0.573	-0.720	-0.874
PdN ₄	-1.132	-1.436	-1.161	-1.096	-1.868	-1.941
PdN ₄	-0.8104	-1.383	-0.913	-0.887	-1.986	-1.790
PdCN ₃	-0.5608	-0.463	-0.563	-0.573	-0.720	-0.877
PdCN ₃	-1.3192	-1.202	-1.270	-1.292	-1.867	-1.590
PdCN ₃	-1.1568	-1.649	-1.128	-1.091	-1.988	-2.150
AgN ₄	-0.6	0.324	-0.596	-0.583	-0.557	-0.137
AgN ₄	-1.1088	-0.656	-1.102	-1.073	-1.461	-1.130
AgCN ₃	-0.5856	0.338	-0.590	-0.564	-0.557	-0.127
AgCN ₃	-1.0744	-0.555	-1.113	-1.075	-1.375	-1.072
AgCN ₃	-1.0728	-0.967	-1.112	-1.125	-1.461	-1.412
SnN ₄	-0.3696	-0.751	-0.414	-0.429	-0.595	-0.965
SnN ₄	-1.0072	-1.118	-1.666	-1.037	-1.544	-2.104
SnN ₄	-1.5736	-1.614	-1.456	-1.523	-1.640	-1.945
SnCN ₃	-0.4656	-0.726	-1.456	-0.484	-0.596	-1.289
SnCN ₃	-1.4432	-1.996	-1.455	-1.423	-1.544	-2.231
SnCN ₃	-1.2	-1.610	-1.315	-1.229	-1.638	-1.576
WN ₄	-1.7816	-2.195	-1.898	-1.819	-0.792	-1.892
WN ₄	-3.58	-2.916	-3.627	-3.616	-1.906	-2.706
WN ₄	-3.6336	-3.508	-3.642	-3.649	-2.007	-2.936
WCN ₃	-2.8208	-2.289	-2.349	-2.801	-0.792	-1.856
WCN ₃	-3.7168	-3.397	-3.645	-3.657	-1.906	-2.767

WCN ₃	-3.8864	-3.078	-3.803	-3.851	-2.007	-2.890
IrN ₄	-0.54	-0.515	-0.543	-0.552	-0.640	-0.655
IrN ₄	-1.3256	-1.633	-1.567	-1.416	-1.439	-1.820
IrN ₄	-1.6512	-1.405	-1.530	-1.620	-1.507	-1.481
IrCN ₃	-0.6048	-0.917	-0.795	-0.668	-0.641	-1.001
IrCN ₃	-1.9784	-1.715	-2.013	-1.874	-1.439	-1.838
IrCN ₃	-2.0504	-1.846	-1.981	-2.023	-1.508	-2.012
PtN ₄	-0.5528	-0.414	-0.556	-0.568	-0.602	-0.251
PtN ₄	-1.12	-1.432	-1.144	-1.101	-1.090	-1.358
PtN ₄	-0.8208	-1.354	-0.912	-0.893	-1.128	-1.182
PtCN ₃	-0.5584	-0.424	-0.557	-0.568	-0.602	-0.261
PtCN ₃	-1.3096	-1.178	-1.256	-1.305	-1.089	-0.990
PtCN ₃	-1.1392	-1.639	-1.113	-1.096	-1.128	-1.567
AuN ₄	-0.5872	-0.106	-0.584	-0.553	-0.610	-0.143
AuN ₄	-1.0728	-0.907	-1.091	-1.067	-1.000	-0.959
AuN ₄	-0.8712	-1.208	-0.925	-0.984	-1.029	-1.320
AuCN ₃	-0.58	-0.112	-0.581	-0.553	-0.610	-0.148
AuCN ₃	-1.1	-0.916	-1.090	-1.067	-1.000	-0.967
AuCN ₃	-0.9288	-1.199	-0.928	-0.984	-1.029	-1.314
BiN ₄	-0.5	-0.413	-0.475	-0.473	-0.774	-0.292
BiN ₄	-1.4152	-1.940	-1.526	-1.481	-1.936	-1.603
BiCN ₃	-0.4464	-0.450	-0.461	-0.473	-0.774	-0.304
BiCN ₃	-1.836	-2.309	-1.680	-1.806	-1.936	-1.749

Figure S4. Average MSE/RMSE values of LR, RFR, and GBR over the same 20 repeated and randomized data, respectively.

Table S5. The average MSE/RMSE value of LR, RFR and GBR over the same 20 repeated and randomized data.

	MSE	RMSE
LR	0.244	0.490
RFR	0.077	0.271
GBR	0.070	0.259

After analyzing the results, the predicted performance of the five ML algorithms was evaluated, in which SVR and KRR algorithms show distinct overfitting with relatively large or zero RMSE (Figure S3a and S3b). Consequently, the other three algorithms were mainly used to observe the fitting results, and it was found that GBR algorithm (vs. RFR and LR algorithm, Figure 1e, Figure S2a, S2b and Table S3, Table S4) gives a much better fitting. Then, the dataset was split into training and learning sets over the same 20 repeated and randomized data for ML prediction. Similar with fitting results, LR algorithm shows a poor prediction, with the R^2 and RMSE values of the training/testing sets are 0.759 and 0.435 eV, respectively (Figure S4a). The testing set with RFR algorithm exhibits a better prediction performance than the training set, with the R^2 of 0.969 and RMSE of 0.156 eV (Figure S4b). It may indicate that the predicted Eds values present a deviation from the actual Eds values and prediction of RFR overestimated Eds. Corresponding to a better fitting result, GBR algorithm presents a relatively accurate prediction with the R^2 and RMSE values of the training/testing sets are 0.97 and 0.153 eV, respectively (Figure S4c). The average errors of RMSE and MSE for LR, RFR and GBR algorithms over 20 times of random training and learning are shown in Figure S5 and Table S5. Both RFR and GBR algorithms give a lower error bar of RMSE/MSE than LR algorithm, consistent with the model prediction results. According to the prediction in Figure 1e, a linear relationship between the adsorption energy Eads and diverse SACs is obtained, which demonstrates that the prediction results of ML are like those of DFT calculations. Meanwhile, compared to DFT calculations, machine learning can accelerate the collection of large numbers of computational results.

(3) Please clarify the size and structure of the data used for ML training. Is it only 69 simulation cases?

Response: Thank you for the reviewer's question. Yes, there are 69 simulation cases in the first draft. The reason that the number of simulation cases is lower than the typical ML training is the limited metal elements that can construct a M-N₄ structure. To increase the generality of our simulation, we have extended the simulation cases from 69 to 123 DFT calculation datasets by adopting the M-N₄ and M-C₁N₃ structure in the revised manuscript. We understand that machine learning typically benefits from larger datasets, and we have made efforts to expand the DFT calculation dataset in the last two months. We excluded metal types that could not reliably form the M-N₄ and M-C₁N₃ structure, resulting in a suitable database that we believe provides a reasonable input for enhancing the accuracy of analysis and predictions. The changes made in this work can be found in the response to the Comment #2.

(4) More details are needed to evaluate the authors' work on ML. It is currently not convincing at all.

Response: We really appreciate this comment from the reviewer. To make this work more convinced, we supplemented more DFT data as discussed above. At the same time, we changed the k value and the training/testing samples, increasing the test set. In addition, to emphasize that ML is one of the steps in predicting efficient catalysts, we adjust the ML part in the article, and more details about the ML procedures have been moved in the supplementary information. Just a quick note to emphasize the importance of this work. Although we can not provide datasets as many as a typical ML simulation, we have tried our best to construct DFT models by using the only 19 metals that are able to form single atoms. This work is the first one to build scaling relationship between single atoms catalysts and product selectivity in sodium-sulfur batteries. This work aims to inspire future research in the field of sodium-sulfur batteries, and we believe that upcoming work will further enhance machine learning models and data volume.

Revision made:

(Manuscript, page 6)

Next, machine learning is performed as a sufficient way to identify the best scientific descriptors, which can assess how various physical and chemical properties of metal atoms affect adsorption or reaction energy. As shown in Figure S1, it is obviously that the bond length of metal and sulfur can influence the adsorption and reaction energy. According to the prediction in Figure 1e, a linear relationship between the adsorption energy E_{ads} and diverse SACs is obtained, which demonstrates that the prediction results of ML are like those of DFT calculations. Then, by establishing a linear relationship between the lengths of adsorption and metal-sulfur bonds (Figure 1d), combined with the theoretical knowledge discussion, we can rapidly screen potential SACs for Na-S batteries that should exist in a specific region (as shown as Figure 1d, optimized region), in which the SACs, Mn-N₄, Fe-N₄, Rh-N₄, Mg-N₄, Co-N₄ and Mg-C₁N₃ feature mild adsorption (more detailed machine learning procedures are in the Supplementary information, Figure S2, Figure S3, Figure S4 and Table S2, Table S3, Table S4, Table S5). With the assistance of prediction, we selected six representative SACs, Mn₁, Fe₁, Co₁, Sn₁, Cu₁, and Ni₁, which were used to conduct further experimental validation in Na-S batteries.

(Supplementary information, Experimental section)

Machine Learning (ML) Methods. A method combining density functional theory (DFT) calculations with machine learning (ML) by using the bond length of M-S and the adsorption energies of the metal with sulfur, Na₂S, and Na₂S₄ as indicators to predict advanced single atom catalysts for high-performance RT Na-S batteries. Here, a total of 123 adsorption energies of different MN₄ and MC₁N₃ sites were obtained by DFT calculations and 123 available DFT data (with the consideration of the stability of materials and thereby deleting the data with broken structures) were used for machine training and learning in five ML models (i.e. linear regression (LR), random forest regression (RFR), gradient boosted regression (GBR), support vector regression (SVR), and Kernel ridge regression (KRR) algorithms) coupling with the elemental information. The relationships between the used descriptors and adsorption activity were firstly analyzed through the Pearson correlation coefficient (Pearson), the Spearman correlation coefficient (Spearman) and the RandomForest feature importance (RandomForest) methods, as shown in Figure S1 and Table S2. The rankings of these descriptors by Pearson are basically consistent with that by Spearman, except for the metal-sulfur bond (Lms), whereas RandomForest method displays obviously distinction although the top two rankings are same with Spearman method. The rankings indicate that adsorption energies to polysulfides and sodium sulfide exhibits low correlations with most descriptors except the change of metal-nitrogen bond length (D Lmn), Lms and valence electron number of metal element (Nve) and adsorbed species (Ms). After a rough screening of features according to the degree of feature importance, 11 main features (Figure S1) were chosen as descriptors. The *sklearn* package was employed to implement the data processing and import the ML algorithms. After analyzing the importance of features, the predicted performance of the five ML algorithms was evaluated, in which SVR and KRR algorithms show distinct overfitting with relatively large or zero RMSE (Figure S3a and S3b). Consequently, the other three algorithms were mainly used to observe the fitting results, and it was found that GBR algorithm (vs. RFR and LR algorithm, Figure 1e, Figure S2a, S2b and Table S3, Table S4) gives a much better fitting. Then, the dataset was split into training and learning sets over the same 20 repeated and randomized data for ML prediction. Similar with fitting results, LR algorithm shows a poor prediction, with the R² and RMSE values of the training/testing sets are 0.759 and 0.435 eV, respectively (Supplementary information, Figure S4a). The testing set with RFR algorithm exhibits a better prediction performance than the training set, with the R² of 0.969 and RMSE of 0.156 eV (Supplementary information, Figure S4b). It may indicate that the predicted Eds values present a deviation from the actual Eds values and prediction of RFR overestimated Eds. Corresponding to a better fitting result, GBR algorithm presents a relatively accurate prediction with the R² and RMSE values of the training/testing sets are 0.97 and 0.153 eV, respectively

(Supplementary information, Figure S4c). The average errors of RMSE and MSE for LR, RFR and GBR algorithms over 20 times of random training and learning are shown in Figure S5 and Table S5. Both RFR and GBR algorithms give a lower error bar of RMSE/MSE than LR algorithm, consistent with the model prediction results. According to the prediction in Figure 1e, a linear relationship between the adsorption energy E_{ads} and diverse SACs is obtained, which demonstrates that the prediction results of ML are like those of DFT calculations. Meanwhile, compared to DFT calculations, machine learning can accelerate the collection of large numbers of computational results.

LR: linear regression; RFR: random forest regression; GBR: gradient boosted regression; SVR: support vector regression; KRR: Kernel ridge regression

Table S1. List of DFT-calculated and elemental features used as descriptors.

Features	
Period	Period number of metal element
RWIGS	Bulk wigner-seitz radius of metal element
Rm	Atomic radius
Nve	Valence electron number of metal element
Am	Electron affinity
Mm	Atomic mass of metal element
Xm	Electronegativity of metal element
Dm	Density of metal element
Ms	Atomic mass of adsorption species
Lms	Length of metal-sulfur bond
D Lmn	Average change of metal-nitrogen bond length

Figures and Tables

Table S2. Values of important 11 features through the RandomForest method over the 10 randomized data, and Pearson and Spearman methods.

	Pearson	Spearman	RandomForest
Period	0.077	0.15	0.005
Group	0.369	0.394	0.142
RWIGS	0.074	0.111	0.024
Nve	0.389	0.405	0.089
Am	0.22	0.293	0.019
Mm	0.118	0.278	0.016
Xm	0.052	0.137	0.015
Dm	0.093	0.163	0.014
Ms	0.577	0.58	0.203
Lms	0.502	0.639	0.412

Figure S1. Importance of used 11 features through the RandomForest method over the 10 randomized data, and Pearson and Spearman methods.

Figure S2. Comparison of the adsorption energy (E_{ds}) from the DFT calculations and the full-fit results using the various ML algorithms: (a) RFR and (b) LR, respectively. R^2 : coefficient of determination; RMSE: root mean square error.

Figure S3. Predicted vs. DFT-calculated E_{ds} of single-atom catalyst (SAC) materials by using SVR (a), and KRR (b) algorithms, with 80% training (blue dot) and 20% testing (red dot) data.

Table S3. Feature values for metal elements in ML modelling, including atomic number of metal element, period number of metal element, the group of metal element, bulk wigner-seitz radius of metal element, valence electron number of metal element, atomic mass of metal element, electronegativity of metal element, density of metal element.

Metal	Period	Group	RWIGS	Nve	Am	Mm	Xm	Dm	Ms	Lms	D Lmn
MgN ₄	3	2	2.88	2	-40	24.03	1.31	1.74	256.48	2.773	-0.03
MgN ₄	3	2	2.88	2	-40	24.03	1.31	1.74	110.12	2.402	-0.126
MgN ₄	3	2	2.88	2	-40	24.03	1.31	1.74	78.06	2.62	-0.112
MgCN ₃	3	2	2.88	2	-40	24.03	1.31	1.74	256.48	2.705	-0.046
MgCN ₃	3	2	2.88	2	-40	24.03	1.31	1.74	110.12	2.511	-0.105
MgCN ₃	3	2	2.88	2	-40	24.03	1.31	1.74	78.06	2.414	-0.159
AlN ₄	3	13	2.65	3	41.76	26.98	1.61	2.7	256.48	2.621	-0.017
AlN ₄	3	13	2.65	3	41.76	26.98	1.61	2.7	110.12	2.24	-0.084
AlN ₄	3	13	2.65	3	41.76	26.98	1.61	2.7	78.06	2.322	-0.058
AlCN ₃	3	13	2.65	3	41.76	26.98	1.61	2.7	256.48	2.653	-0.011
AlCN ₃	3	13	2.65	3	41.76	26.98	1.61	2.7	110.12	2.343	-0.055
AlCN ₃	3	13	2.65	3	41.76	26.98	1.61	2.7	78.06	2.261	-0.084
SiN ₄	3	14	2.48	4	134.06	28.08	1.9	2.33	256.48	2.184	-0.071
SiN ₄	3	14	2.48	4	134.06	28.08	1.9	2.33	110.12	2.128	-0.078
SiN ₄	3	14	2.48	4	134.06	28.08	1.9	2.33	78.06	2.213	-0.057
SiCN ₃	3	14	2.48	4	134.06	28.08	1.9	2.33	256.48	2.657	-0.008
SiCN ₃	3	14	2.48	4	134.06	28.08	1.9	2.33	110.12	2.228	-0.052
SiCN ₃	3	14	2.48	4	134.06	28.08	1.9	2.33	78.06	2.148	-0.073
TiN ₄	4	4	2.5	4	7.28	47.87	1.54	4.51	256.48	2.43	-0.032
TiN ₄	4	4	2.5	4	7.28	47.87	1.54	4.51	110.12	2.261	0.028
TiN ₄	4	4	2.5	4	7.28	47.87	1.54	4.51	78.06	2.414	0.017
TiCN ₃	4	4	2.5	4	7.28	47.87	1.54	4.51	256.48	2.355	-0.01

TiCN ₃	4	4	2.5	4	7.28	47.87	1.54	4.51	110.12	2.412	-0.035
TiCN ₃	4	4	2.5	4	7.28	47.87	1.54	4.51	78.06	2.27	-0.018
MnN ₄	4	7	2.5	7	-50	54.94	1.55	7.21	256.48	2.656	-0.008
MnN ₄	4	7	2.5	7	-50	54.94	1.55	7.21	110.12	2.3	-0.042
MnN ₄	4	7	2.5	7	-50	54.94	1.55	7.21	78.06	2.388	-0.023
MnCN ₃	4	7	2.5	7	-50	54.94	1.55	7.21	256.48	2.556	-0.013
MnCN ₃	4	7	2.5	7	-50	54.94	1.55	7.21	110.12	2.28	-0.018
MnCN ₃	4	7	2.5	7	-50	54.94	1.55	7.21	78.06	2.218	-0.032
FeN ₄	4	8	2.46	8	14.78	55.85	1.83	7.87	256.48	2.641	-0.011
FeN ₄	4	8	2.46	8	14.78	55.85	1.83	7.87	110.12	2.213	-0.013
FeN ₄	4	8	2.46	8	14.78	55.85	1.83	7.87	78.06	2.182	-0.013
FeCN ₃	4	8	2.46	8	14.78	55.85	1.83	7.87	256.48	2.338	-0.014
FeCN ₃	4	8	2.46	8	14.78	55.85	1.83	7.87	110.12	2.181	-0.013
FeCN ₃	4	8	2.46	8	14.78	55.85	1.83	7.87	78.06	2.179	-0.024
CoN ₄	4	9	2.46	9	63.89	58.93	1.88	8.86	256.48	2.447	-0.025
CoN ₄	4	9	2.46	9	63.89	58.93	1.88	8.86	110.12	2.258	-0.02
CoN ₄	4	9	2.46	9	63.89	58.93	1.88	8.86	78.06	2.244	-0.022
CoCN ₃	4	9	2.46	9	63.89	58.93	1.88	8.86	256.48	2.26	-0.033
CoCN ₃	4	9	2.46	9	63.89	58.93	1.88	8.86	110.12	2.235	-0.025
CoCN ₃	4	9	2.46	9	63.89	58.93	1.88	8.86	78.06	2.259	-0.023
NiN ₄	4	10	2.43	10	111.65	58.69	1.91	8.9	256.48	3.288	-0.002
NiN ₄	4	10	2.43	10	111.65	58.69	1.91	8.9	110.12	2.502	-0.012
NiN ₄	4	10	2.43	10	111.65	58.69	1.91	8.9	78.06	3.86	0
NiCN ₃	4	10	2.43	10	111.65	58.69	1.91	8.9	256.48	3.22	-0.002
NiCN ₃	4	10	2.43	10	111.65	58.69	1.91	8.9	110.12	3.935	-0.002
NiCN ₃	4	10	2.43	10	111.65	58.69	1.91	8.9	78.06	2.439	-0.019
CuN ₄	4	11	2.2	11	119.23	63.55	1.9	8.96	256.48	3.276	-0.001
CuN ₄	4	11	2.2	11	119.23	63.55	1.9	8.96	110.12	2.502	-0.047
CuN ₄	4	11	2.2	11	119.23	63.55	1.9	8.96	78.06	3.77	0.001
CuCN ₃	4	11	2.2	11	119.23	63.55	1.9	8.96	256.48	3.281	-0.001
CuCN ₃	4	11	2.2	11	119.23	63.55	1.9	8.96	110.12	2.615	-0.023
CuCN ₃	4	11	2.2	11	119.23	63.55	1.9	8.96	78.06	2.471	-0.034
ZnN ₄	4	12	2.4	12	-58	65.38	1.65	7.14	256.48	2.79	-0.019
ZnN ₄	4	12	2.4	12	-58	65.38	1.65	7.14	110.12	2.266	-0.151
ZnN ₄	4	12	2.4	12	-58	65.38	1.65	7.14	78.06	2.327	-0.104
ZnCN ₃	4	12	2.4	12	-58	65.38	1.65	7.14	256.48	3.417	-0.015
ZnCN ₃	4	12	2.4	12	-58	65.38	1.65	7.14	110.12	2.376	-0.104
ZnCN ₃	4	12	2.4	12	-58	65.38	1.65	7.14	78.06	2.278	-0.187
MoN ₄	5	6	2.75	6	72.1	95.96	2.16	10.2 8	256.48	2.2	-0.023

MoN₄	5	6	2.75	6	72.1	95.96	2.16	$\frac{10.2}{8}$	110.12	2.242	-0.027
MoN₄	5	6	2.75	6	72.1	95.96	2.16	$\frac{10.2}{8}$	78.06	2.375	-0.106
MoCN₃	5	6	2.75	6	72.1	95.96	2.16	$\frac{10.2}{8}$	256.48	2.284	-0.077
MoCN₃	5	6	2.75	6	72.1	95.96	2.16	$\frac{10.2}{8}$	110.12	2.359	-0.107
MoCN₃	5	6	2.75	6	72.1	95.96	2.16	$\frac{10.2}{8}$	78.06	2.245	-0.009
RuN₄	5	8	2.65	8	100.27	101.1	2.2	$\frac{12.4}{5}$	256.48	2.136	-0.037
RuN₄	5	8	2.65	8	100.27	101.1	2.2	$\frac{12.4}{5}$	110.12	2.294	-0.009
RuN₄	5	8	2.65	8	100.27	101.1	2.2	$\frac{12.4}{5}$	78.06	2.441	-0.072
RuCN₃	5	8	2.65	8	100.27	101.1	2.2	$\frac{12.4}{5}$	256.48	2.146	-0.049
RuCN₃	5	8	2.65	8	100.27	101.1	2.2	$\frac{12.4}{5}$	110.12	2.331	-0.065
RuCN₃	5	8	2.65	8	100.27	101.1	2.2	$\frac{12.4}{5}$	78.06	2.261	-0.032
RhN₄	5	9	2.65	9	100.27	102.9	2.28	$\frac{12.4}{1}$	256.48	3.078	-0.002
RhN₄	5	9	2.65	9	100.27	102.9	2.28	$\frac{12.4}{1}$	110.12	2.334	-0.018
RhN₄	5	9	2.65	9	100.27	102.9	2.28	$\frac{12.4}{1}$	78.06	4.118	-0.005
RhCN₃	5	9	2.65	9	100.27	102.9	2.28	$\frac{12.4}{1}$	256.48	2.304	-0.038
RhCN₃	5	9	2.65	9	100.27	102.9	2.28	$\frac{12.4}{1}$	110.12	2.319	-0.03
RhCN₃	5	9	2.65	9	100.27	102.9	2.28	$\frac{12.4}{1}$	78.06	2.341	-0.023
PdN₄	5	10	2.71	10	54.24	106.4	2.2	$\frac{12.0}{2}$	256.48	3.486	-0.003
PdN₄	5	10	2.71	10	54.24	106.4	2.2	$\frac{12.0}{2}$	110.12	2.728	-0.006
PdN₄	5	10	2.71	10	54.24	106.4	2.2	$\frac{12.0}{2}$	78.06	3.95	-0.002
PdCN₃	5	10	2.71	10	54.24	106.4	2.2	$\frac{12.0}{2}$	256.48	3.477	-0.003
PdCN₃	5	10	2.71	10	54.24	106.4	2.2	$\frac{12.0}{2}$	110.12	4.002	-0.004
PdCN₃	5	10	2.71	10	54.24	106.4	2.2	$\frac{12.0}{2}$	78.06	2.659	-0.01
AgN₄	5	11	2.84	11	125.86	107.9	1.93	$\frac{10.4}{9}$	256.48	3.443	-0.005

AgN ₄	5	11	2.84	11	125.86	107.9	1.93	$\frac{10.4}{9}$	78.06	3.63	-0.006
AgCN ₃	5	11	2.84	11	125.86	107.9	1.93	$\frac{10.4}{9}$	256.48	3.472	-0.003
AgCN ₃	5	11	2.84	11	125.86	107.9	1.93	$\frac{10.4}{9}$	110.12	3.154	-0.005
AgCN ₃	5	11	2.84	11	125.86	107.9	1.93	$\frac{10.4}{9}$	78.06	2.694	-0.037
SnN ₄	5	14	2.96	4	107.29	118.7	1.96	7.26	256.48	3.866	0.002
SnN ₄	5	14	2.96	4	107.29	118.7	1.96	7.26	110.12	2.378	0.156
SnN ₄	5	14	2.96	4	107.29	118.7	1.96	7.26	78.06	4.024	0.027
SnCN ₃	5	14	2.96	4	107.29	118.7	1.96	7.26	256.48	2.516	0.058
SnCN ₃	5	14	2.96	4	107.29	118.7	1.96	7.26	110.12	2.481	-0.035
SnCN ₃	5	14	2.96	4	107.29	118.7	1.96	7.26	78.06	5.535	-0.029
WN ₄	6	6	2.75	6	78.76	183.8	2.36	19.3	256.48	2.139	-0.055
WN ₄	6	6	2.75	6	78.76	183.8	2.36	19.3	110.12	2.236	-0.036
WN ₄	6	6	2.75	6	78.76	183.8	2.36	19.3	78.06	2.355	-0.128
WCN ₃	6	6	2.75	6	78.76	183.8	2.36	19.3	256.48	2.353	-0.083
WCN ₃	6	6	2.75	6	78.76	183.8	2.36	19.3	110.12	2.328	-0.142
WCN ₃	6	6	2.75	6	78.76	183.8	2.36	19.3	78.06	2.239	-0.032
IrN ₄	6	9	2.84	9	150.94	192.2	2.2	$\frac{22.5}{6}$	256.48	3.392	-0.002
IrN ₄	6	9	2.84	9	150.94	192.2	2.2	$\frac{22.5}{6}$	110.12	2.331	-0.024
IrN ₄	6	9	2.84	9	150.94	192.2	2.2	$\frac{22.5}{6}$	78.06	4.2	-0.007
IrCN ₃	6	9	2.84	9	150.94	192.2	2.2	$\frac{22.5}{6}$	256.48	2.259	-0.045
IrCN ₃	6	9	2.84	9	150.94	192.2	2.2	$\frac{22.5}{6}$	110.12	2.316	-0.041
IrCN ₃	6	9	2.84	9	150.94	192.2	2.2	$\frac{22.5}{6}$	78.06	2.337	-0.031
PtN ₄	6	10	2.75	10	205.04	195.1	2.28	$\frac{21.4}{5}$	256.48	3.584	-0.002
PtN ₄	6	10	2.75	10	205.04	195.1	2.28	$\frac{21.4}{5}$	110.12	2.69	-0.009
PtN ₄	6	10	2.75	10	205.04	195.1	2.28	$\frac{21.4}{5}$	78.06	3.999	-0.003
PtCN ₃	6	10	2.75	10	205.04	195.1	2.28	$\frac{21.4}{5}$	256.48	3.552	-0.003
PtCN ₃	6	10	2.75	10	205.04	195.1	2.28	$\frac{21.4}{5}$	110.12	4.02	-0.005
PtCN ₃	6	10	2.75	10	205.04	195.1	2.28	$\frac{21.4}{5}$	78.06	2.621	-0.012
AuN ₄	6	11	2.84	11	222.75	197	2.54	$\frac{19.3}{2}$	256.48	3.542	-0.002

AuN ₄	6	11	2.84	11	222.75	197	2.54	$\frac{19.3}{2}$	110.12	3.69	-0.002
AuN ₄	6	11	2.84	11	222.75	197	2.54	$\frac{19.3}{2}$	78.06	3.063	-0.004
AuCN ₃	6	11	2.84	11	222.75	197	2.54	$\frac{19.3}{2}$	256.48	3.53	-0.003
AuCN ₃	6	11	2.84	11	222.75	197	2.54	$\frac{19.3}{2}$	110.12	3.664	-0.003
AuCN ₃	6	11	2.84	11	222.75	197	2.54	$\frac{19.3}{2}$	78.06	3.084	-0.003
BiN ₄	6	15	3.09	5	90.92	209	2.02	9.78	256.48	3.746	0.003
BiN ₄	6	15	3.09	5	90.92	209	2.02	9.78	78.06	3.015	-0.08
BiCN ₃	6	15	3.09	5	90.92	209	2.02	9.78	256.48	3.721	-0.004
BiCN ₃	6	15	3.09	5	90.92	209	2.02	9.78	78.06	2.679	-0.146

Table S4. Eds from the DFT calculations and the full-fit results using the three ML algorithms, LR, RFR and GBR, respectively.

Metal	Eds	LR-pred	RFR-pred	GBR-pred	SVR-pred	KRR-pred
MgN ₄	-0.684	-0.732	-0.702	-0.714	-0.793	-0.997
MgN ₄	-1.6104	-2.075	-1.841	-1.689	-1.869	-2.033
MgN ₄	-1.9576	-2.153	-1.888	-1.894	-2.004	-2.149
MgCN ₃	-0.72	-0.820	-0.736	-0.713	-0.793	-1.028
MgCN ₃	-1.8096	-1.958	-1.811	-1.793	-1.869	-1.986
MgCN ₃	-1.8728	-2.409	-1.939	-1.899	-2.004	-2.244
AlN ₄	-0.7832	-0.801	-0.755	-0.713	-0.841	-1.006
AlN ₄	-2.2632	-2.010	-2.330	-2.289	-1.962	-2.021
AlN ₄	-2.624	-2.055	-2.453	-2.554	-2.080	-2.165
AlCN ₃	-0.7328	-0.768	-0.735	-0.707	-0.841	-0.993
AlCN ₃	-2.1344	-1.856	-2.294	-2.236	-1.962	-1.970
AlCN ₃	-2.344	-2.188	-2.358	-2.443	-2.080	-2.203
SiN ₄	-1.0936	-1.366	-1.370	-1.134	-0.749	-0.990
SiN ₄	-2.6448	-2.236	-2.689	-2.626	-1.445	-1.867
SiN ₄	-3.1856	-2.303	-2.915	-3.062	-1.513	-2.015
SiCN ₃	-0.68	-0.987	-0.675	-0.699	-0.749	-0.809
SiCN ₃	-2.44	-2.096	-2.579	-2.487	-1.444	-1.819
SiCN ₃	-2.7984	-2.390	-2.797	-2.809	-1.513	-2.045
TiN ₄	-2.1176	-1.980	-1.828	-2.084	-0.817	-1.477
TiN ₄	-3.2048	-2.555	-3.172	-3.238	-1.997	-2.331
TiN ₄	-3.556	-2.761	-3.263	-3.456	-2.128	-2.486
TiCN ₃	-1.4496	-1.890	-1.608	-1.535	-0.817	-1.480

TiCN ₃	-3.2888	-2.824	-3.124	-3.196	-1.997	-2.341
TiCN ₃	-3.4976	-2.951	-3.329	-3.395	-2.128	-2.554
MnN ₄	-1.014	-1.061	-0.764	-1.104	-0.751	-1.082
MnN ₄	-1.6904	-2.111	-1.882	-1.827	-1.818	-2.063
MnN ₄	-1.8976	-2.187	-1.949	-1.937	-1.955	-2.211
MnCN ₃	-0.7032	-1.103	-0.721	-0.730	-0.751	-1.114
MnCN ₃	-1.8312	-2.002	-1.890	-1.780	-1.818	-2.049
MnCN ₃	-2.172	-2.260	-2.122	-2.169	-1.955	-2.265
FeN ₄	-0.6936	-1.035	-0.705	-0.730	-0.794	-1.028
FeN ₄	-1.8984	-1.947	-1.892	-1.906	-1.975	-2.003
FeN ₄	-1.8032	-2.133	-1.852	-1.893	-2.106	-2.200
FeCN ₃	-0.728	-1.102	-0.950	-0.778	-0.794	-1.114
FeCN ₃	-1.9608	-1.952	-1.921	-1.904	-1.975	-2.012
FeCN ₃	-2.1496	-2.185	-2.048	-2.120	-2.106	-2.209
CoN ₄	-0.7152	-0.843	-0.713	-0.707	-0.750	-0.830
CoN ₄	-1.6528	-1.680	-1.673	-1.582	-1.830	-1.733
CoN ₄	-1.5288	-1.873	-1.620	-1.650	-1.944	-1.926
CoCN ₃	-0.7128	-0.914	-0.805	-0.725	-0.751	-0.887
CoCN ₃	-1.8872	-1.708	-1.842	-1.778	-1.830	-1.744
CoCN ₃	-1.7408	-1.875	-1.762	-1.799	-1.944	-1.923
NiN ₄	-0.568	-0.381	-0.572	-0.568	-0.671	-0.314
NiN ₄	-1.1848	-1.394	-1.213	-1.085	-1.516	-1.394
NiN ₄	-0.9792	-1.279	-1.013	-1.065	-1.603	-1.199
NiCN ₃	-0.5712	-0.393	-0.573	-0.568	-0.671	-0.333
NiCN ₃	-1.3448	-1.094	-1.268	-1.253	-1.515	-0.993
NiCN ₃	-1.1608	-1.619	-1.402	-1.424	-1.604	-1.604
CuN ₄	-0.588	-0.594	-0.592	-0.594	-0.641	-0.045
CuN ₄	-1.204	-1.773	-1.212	-1.204	-1.445	-1.151
CuN ₄	-1.12	-1.505	-1.055	-1.076	-1.528	-0.952
CuCN ₃	-0.5976	-0.593	-0.596	-0.594	-0.641	-0.044
CuCN ₃	-1.3448	-1.641	-1.277	-1.244	-1.445	-1.101
CuCN ₃	-1.2032	-1.899	-1.199	-1.231	-1.529	-1.337
ZnN ₄	-0.6288	-0.034	-0.635	-0.618	-0.729	-0.149
ZnN ₄	-1.7616	-1.573	-1.826	-1.761	-1.763	-1.256
ZnN ₄	-1.9504	-1.523	-1.896	-1.910	-1.899	-1.389
ZnCN ₃	-0.6488	0.095	-0.630	-0.663	-0.729	0.026
ZnCN ₃	-1.6904	-1.333	-1.797	-1.700	-1.763	-1.188
ZnCN ₃	-1.7992	-1.921	-1.863	-1.834	-1.899	-1.470
MoN ₄	-1.4016	-1.753	-1.556	-1.433	-0.706	-1.919

MoN ₄	-3.0064	-2.592	-3.057	-3.130	-1.793	-2.767
MoN ₄	-3.3312	-3.120	-3.309	-3.306	-1.905	-2.982
MoCN ₃	-1.9824	-1.992	-1.979	-1.930	-0.706	-1.940
MoCN ₃	-3.1168	-2.947	-3.135	-3.163	-1.793	-2.800
MoCN ₃	-3.432	-2.688	-3.306	-3.311	-1.905	-2.939
RuN ₄	-1.2136	-1.506	-1.181	-1.227	-0.634	-1.598
RuN ₄	-2.032	-2.174	-2.119	-2.069	-1.593	-2.388
RuN ₄	-2.484	-2.625	-2.070	-2.537	-1.692	-2.587
RuCN ₃	-1.4832	-1.561	-1.328	-1.397	-0.634	-1.605
RuCN ₃	-2.5888	-2.430	-2.298	-2.497	-1.593	-2.424
RuCN ₃	-2.8	-2.469	-2.593	-2.626	-1.692	-2.604
RhN ₄	-0.5304	-0.960	-0.539	-0.570	-0.627	-1.187
RhN ₄	-1.3616	-1.993	-1.531	-1.576	-1.587	-2.260
RhN ₄	-1.6704	-1.799	-1.537	-1.645	-1.685	-1.948
RhCN ₃	-0.6616	-1.265	-0.777	-0.716	-0.627	-1.429
RhCN ₃	-1.9968	-2.052	-1.958	-1.960	-1.587	-2.274
RhCN ₃	-2.1048	-2.197	-1.929	-1.873	-1.686	-2.450
PdN ₄	-0.5568	-0.462	-0.561	-0.573	-0.720	-0.874
PdN ₄	-1.132	-1.436	-1.161	-1.096	-1.868	-1.941
PdN ₄	-0.8104	-1.383	-0.913	-0.887	-1.986	-1.790
PdCN ₃	-0.5608	-0.463	-0.563	-0.573	-0.720	-0.877
PdCN ₃	-1.3192	-1.202	-1.270	-1.292	-1.867	-1.590
PdCN ₃	-1.1568	-1.649	-1.128	-1.091	-1.988	-2.150
AgN ₄	-0.6	0.324	-0.596	-0.583	-0.557	-0.137
AgN ₄	-1.1088	-0.656	-1.102	-1.073	-1.461	-1.130
AgCN ₃	-0.5856	0.338	-0.590	-0.564	-0.557	-0.127
AgCN ₃	-1.0744	-0.555	-1.113	-1.075	-1.375	-1.072
AgCN ₃	-1.0728	-0.967	-1.112	-1.125	-1.461	-1.412
SnN ₄	-0.3696	-0.751	-0.414	-0.429	-0.595	-0.965
SnN ₄	-1.0072	-1.118	-1.666	-1.037	-1.544	-2.104
SnN ₄	-1.5736	-1.614	-1.456	-1.523	-1.640	-1.945
SnCN ₃	-0.4656	-0.726	-1.456	-0.484	-0.596	-1.289
SnCN ₃	-1.4432	-1.996	-1.455	-1.423	-1.544	-2.231
SnCN ₃	-1.2	-1.610	-1.315	-1.229	-1.638	-1.576
WN ₄	-1.7816	-2.195	-1.898	-1.819	-0.792	-1.892
WN ₄	-3.58	-2.916	-3.627	-3.616	-1.906	-2.706
WN ₄	-3.6336	-3.508	-3.642	-3.649	-2.007	-2.936
WCN ₃	-2.8208	-2.289	-2.349	-2.801	-0.792	-1.856
WCN ₃	-3.7168	-3.397	-3.645	-3.657	-1.906	-2.767

WCN ₃	-3.8864	-3.078	-3.803	-3.851	-2.007	-2.890
IrN ₄	-0.54	-0.515	-0.543	-0.552	-0.640	-0.655
IrN ₄	-1.3256	-1.633	-1.567	-1.416	-1.439	-1.820
IrN ₄	-1.6512	-1.405	-1.530	-1.620	-1.507	-1.481
IrCN ₃	-0.6048	-0.917	-0.795	-0.668	-0.641	-1.001
IrCN ₃	-1.9784	-1.715	-2.013	-1.874	-1.439	-1.838
IrCN ₃	-2.0504	-1.846	-1.981	-2.023	-1.508	-2.012
PtN ₄	-0.5528	-0.414	-0.556	-0.568	-0.602	-0.251
PtN ₄	-1.12	-1.432	-1.144	-1.101	-1.090	-1.358
PtN ₄	-0.8208	-1.354	-0.912	-0.893	-1.128	-1.182
PtCN ₃	-0.5584	-0.424	-0.557	-0.568	-0.602	-0.261
PtCN ₃	-1.3096	-1.178	-1.256	-1.305	-1.089	-0.990
PtCN ₃	-1.1392	-1.639	-1.113	-1.096	-1.128	-1.567
AuN ₄	-0.5872	-0.106	-0.584	-0.553	-0.610	-0.143
AuN ₄	-1.0728	-0.907	-1.091	-1.067	-1.000	-0.959
AuN ₄	-0.8712	-1.208	-0.925	-0.984	-1.029	-1.320
AuCN ₃	-0.58	-0.112	-0.581	-0.553	-0.610	-0.148
AuCN ₃	-1.1	-0.916	-1.090	-1.067	-1.000	-0.967
AuCN ₃	-0.9288	-1.199	-0.928	-0.984	-1.029	-1.314
BiN ₄	-0.5	-0.413	-0.475	-0.473	-0.774	-0.292
BiN ₄	-1.4152	-1.940	-1.526	-1.481	-1.936	-1.603
BiCN ₃	-0.4464	-0.450	-0.461	-0.473	-0.774	-0.304
BiCN ₃	-1.836	-2.309	-1.680	-1.806	-1.936	-1.749

Figure S4. Average MSE/RMSE values of LR, RFR, and GBR over the same 20 repeated and randomized data, respectively.

Table S5. The average MSE/RMSE value of LR, RFR and GBR over the same 20 repeated and randomized data.

	MSE	RMSE
LR	0.244	0.490
RFR	0.077	0.271
GBR	0.070	0.259

5. Some axis labels are missing, for example Figure 4(a)(c)(d).

Response: Accordingly, we have added the missed label in Figure 4a. In addition, we added arrows from the surface to the interior in Figure 4c and Figure 4d, indicating that both Na and S are distributed throughout the bulk of the electrode.

Revision made:

(Manuscript, page 12)

Figure 4. Pathway selectivity and cycling stability mechanism. (a) *In-situ* synchrotron-based XRD patterns of S@Mn₁-PNC. (b) *Ex-situ* X-ray absorption spectra of S for S@Mn₁-PNC during the initial cycle. (c) and (d) 3D reconstructed images of TOF-SIMS depth profiles of Na and S after 10 cycles. (e) Normalized depth profiles of secondary ion fragments obtained from the S@Mn₁-PNC electrode after 50 cycles. (f) *k*³-weighted FT-EXAFS curves of S@Mn₁-PNC in R-space during the initial discharge and charge processes. WT plots of the Mn *k*-edge of S@Mn₁-PNC cathode during (g) discharge and (h) charge processes.

REVIEWERS' COMMENTS

Reviewer #1 (Remarks to the Author):

Good revision.

Reviewer #2 (Remarks to the Author):

All my comments were well addressed and I would like to recommend its acceptance.

Reviewer #3 (Remarks to the Author):

The authors have very carefully addressed my previous comments. I think the manuscript is now ready for publication.